# XSKILL: Continual Learning from Experience and Skills in Multimodal Agents

Guanyu Jiang [*1 2]  Zhaochen Su [*1]  Xiaoye Qu [3]  Yi R. (May) Fung [1]

## Abstract

Multimodal agents can now tackle complex reasoning tasks with diverse tools, yet they still suffer from inefficient tool use and inflexible orchestration in open-ended settings. A central challenge is enabling such agents to continually improve without parameter updates by learning from past trajectories. We identify two complementary forms of reusable knowledge essential for this goal: *experiences*, providing concise action-level guidance for tool selection and decision making, and *skills*, providing structured task-level guidance for planning and tool use. To this end, we propose XSKILL, a dual-stream framework for continual learning from experience and skills in multimodal agents. XSKILL grounds both knowledge extraction and retrieval in visual observations. During accumulation, XSKILL distills and consolidates experiences and skills from multipath rollouts via visually grounded summarization and cross-rollout critique. During inference, it retrieves and adapts this knowledge to the current visual context and feeds usage history back into accumulation to form a continual learning loop. Evaluated on five benchmarks across diverse domains with four backbone models, XSKILL consistently and substantially outperforms both tool-only and learning-based baselines. Further analysis reveals that the two knowledge streams play complementary roles in influencing the reasoning behaviors of agents and show superior zero-shot generalization. Code is available at https://github.com/XSkill-Agent/XSkill.

## 1. Introduction

The evolution of Multimodal Large Language Models (MLLMs) has transformed agents from passive perceptual systems into active problem solvers, enabling them to tackle complex reasoning tasks with diverse tools (OpenAI, 2025c; Li et al., 2025d; Su et al., 2025). In open-ended settings, such agents can integrate capabilities spanning visual perception, code execution, and information retrieval. Despite this progress, current multimodal agents still face two fundamental bottlenecks: ❶ **Tool use remains inefficient:** as agents often spend excessive steps on simple problems yet fail to conduct sufficiently deep multi-turn exploration when faced with complex queries (Ashraf et al., 2025; Guo et al., 2025a), a problem that structured *skills* can address by providing reusable workflows and tool templates; ❷ **Tool orchestration remains inflexible:** as most existing systems are confined to single-path execution and show limited ability to compose tools in a manner that generalizes across tasks (Guo et al., 2025b; Hong et al., 2025), a limitation that context-sensitive *experiences* can overcome by capturing tactical knowledge for adaptive tool selection. As foundation models become increasingly capable, an important open question is *how to enable multimodal agents to continually improve their tool-use efficiency and tool-composition flexibility through training-free learning from past trajectories*. However, frozen post-trained backbones still lack an effective mechanism for such continual improvement when faced with unseen tasks or evolving toolsets.

Humans naturally improve their problem-solving ability through continual learning from both experiences and skills. Experiences capture concise and context-sensitive guidance distilled from prior attempts, helping refine local decisions such as tool selection, exploration, and error recovery (Hatalis et al., 2025; Hu et al., 2025; Tang et al., 2025; Cao et al., 2025; Cai et al., 2025). Skills, in contrast, encode more structured and reusable procedures that support higher-level planning and tool orchestration across related tasks (Wang et al., 2024; Zheng et al., 2025a; Wang et al., 2025b; Anthropic, 2026). Taken together, these two forms of knowledge support both efficient execution and flexible problem solving. It is therefore natural to equip multimodal agents with both experiences and skills, allowing them to continually accumulate complementary knowledge for improving tool-use efficiency and tool-composition flexibility in a training-free manner.

However, continual learning with such complementary knowledge remains largely absent from the current literature on multimodal agents. More importantly, existing

*Equal contribution  [1]Hong Kong University of Science and Technology  [2]Zhejiang University  [3]Huazhong University of Science and Technology. Correspondence to: Yi R. (May) Fung <yrfung@ust.hk>.

*Proceedings of the 43$^{rd}$ International Conference on Machine Learning*, Seoul, South Korea. PMLR 306, 2026. Copyright 2026 by the author(s).

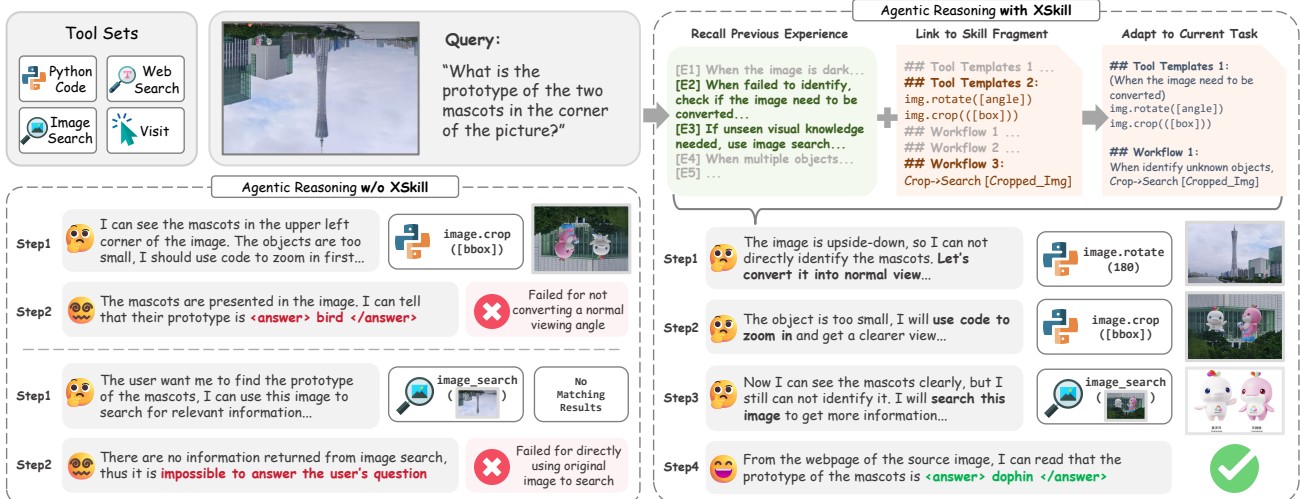

*Figure 1.* Comparison of Reasoning Trajectories on a Multimodal Task with and without XSKILL. The baseline agent (left) fails due to visual-semantic gaps, neglecting to correct the inverted image or isolate small objects. In contrast, XSKILL (right) recalls relevant experiences and links them to structured skill fragments. Through context-aware adaptation, the agent generates a grounded execution plan involving rotation and cropping, leading to successful identification.

approaches rely predominantly on textual trajectory logs during knowledge extraction and retrieval, which is fundamentally inadequate for multimodal settings where critical decision signals are often grounded in visual observations. Without grounding experiences and skills in the visual state of the task, agents cannot reliably retrieve relevant prior knowledge or adapt it to the current execution context. The few related efforts are mostly limited to specialized problems, such as spatial reasoning and GUI navigation (Wu et al., 2025b; Li et al., 2025b), and still fall short of a general solution for multimodal agentic reasoning.

To address these limitations, we propose XSKILL, a dual-stream framework for learning from experience and skills in multimodal agents. XSKILL maintains two complementary forms of knowledge: skills and experiences. Skills provide structured task-level guidance for planning and tool orchestration, while experiences provide concise action-level guidance tied to execution context and failure patterns. The framework then uses these two knowledge streams in a continual learning loop. During the accumulation phase, it extracts skills and experiences from multi-path rollouts through visually grounded summarization and cross-rollout critique, followed by hierarchical consolidation to reduce semantic redundancy. During the inference phase, it retrieves relevant knowledge through task decomposition and adapts it to the current visual context through image-aware rewriting. The resulting usage history is fed back into accumulation, allowing the knowledge base to be progressively refined over time. Although our experiments demonstrate this loop in a single accumulation-then-test cycle, the architecture is designed to support iterative refinement as additional tasks are encountered. As illustrated in Figure 1,

this design enables XSKILL to convert recalled experience and linked skill fragments into a grounded execution plan for visual tasks that remain challenging for strong baselines.

We evaluate XSKILL on diverse multimodal benchmarks spanning visual agentic tool use, multimodal search, and comprehensive multimodal reasoning. Across four backbone models, XSKILL consistently outperforms strong baselines and generalizes effectively to unseen tasks, improving Average@4 by 2.58 to 6.71 points over the tool-only baseline across models, with gains of up to 11.13 points over the strongest baseline on challenging settings. Our contributions are summarized as follows:

- We propose XSKILL. To the best of our knowledge, it is the first framework that unifies visually-grounded task-level skills and action-level experiences in a dual-stream design for multimodal agents, enabling training-free knowledge accumulation from visual agentic tasks.

- We demonstrate consistent improvements over strong baselines across diverse multimodal benchmarks and backbone models, showing that continual learning from experiences and skills substantially enhances multimodal agent performance.

- We provide extensive analysis of robustness and generalization, showing that the two knowledge streams play complementary roles in improving tool-use robustness and enabling stronger zero-shot cross-task transfer.

## 2. Methodology

In this section, we present XSKILL, which structures knowledge into a dual-layer representation to address the bottle-

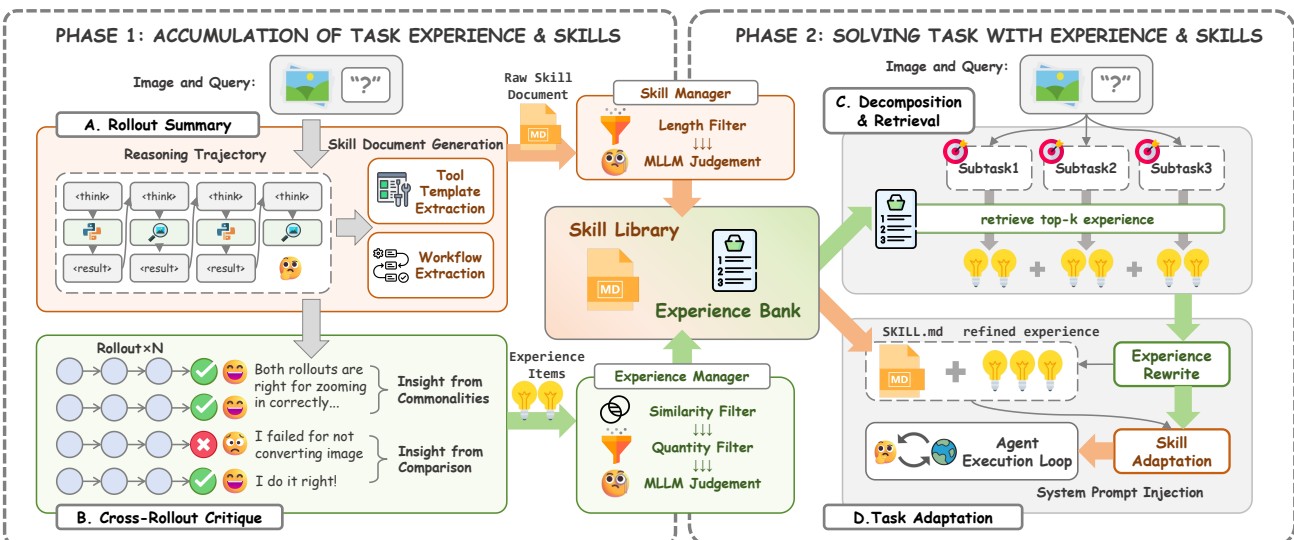

*Figure 2.* Overview of the XSKILL framework. **Phase I** (left): The agent accumulates knowledge by distilling structured skill documents (orange dataflow) and experience items (green dataflow) from multi-path trajectories through **(A) Rollout Summary**, **(B) Cross-Rollout Critique**, and hierarchical consolidation. **Phase II** (right): For a test task, the system **(C) decomposes it into subtasks and retrieves** relevant knowledge, **(D) adapts** it to the current visual context, and injects it into the prompt of the agent for execution.

necks of inefficient tool use and inflexible orchestration. **Skills** provide task-level guidance through structured workflows, while **Experiences** offer action-level insights for specific execution contexts. These are stored in a Markdown-based Skill Library $\mathcal{K}$ and a JSON-based Experience Bank $\mathcal{E}$, respectively. We employ two specialized MLLM instances: $\text{MLLM}_{\text{exec}}$ performs tool-use inference, while $\text{MLLM}_{\text{kb}}$ handles knowledge base operations including extraction, consolidation, and adaptation. This separation allows using a stronger model for knowledge management while keeping the execution model flexible, and enables cross-model knowledge transfer where knowledge accumulated by one model can benefit another.

The architecture of XSKILL consists of two phases (see Figure 2). In Phase I (accumulation), given a training dataset $\mathcal{D}_{\text{train}}$, the agent performs multiple rollouts and employs visually-grounded summarization and cross-rollout critique to extract skills and experiences. In Phase II (inference), for a test task $\mathcal{T}_{\text{test}}$, the system decomposes the query into subtasks, retrieves relevant knowledge via semantic similarity, and adapts it to the current visual context before injection into the system prompt.

### 2.1. Problem Formulation

To capture insights from multimodal trajectories, we separate task-level and action-level knowledge into two complementary structures:

**Definition 2.1** (**Skill**). A *Skill* $k \in \mathcal{K}$ is a task-level guidance document that provides structured workflows for a specific class of problems. Formally, $k = (\mathcal{M}, \mathcal{W}, \mathcal{P})$, where $\mathcal{M}$

denotes metadata consisting of skill name, description, and version number; $\mathcal{W}$ is the workflow sequence, and $\mathcal{P}$ represents reusable tool templates. It is stored as a structured Markdown document.

**Definition 2.2** (**Experience**). An *Experience* $e \in \mathcal{E}$ serves as a non-parametric tactical prompt, capturing action-level insights that are often omitted in high-level instructions. Each experience is structured as $e = (c, a, \mathbf{v}_e)$, where $c$ describes the triggering condition, $a$ provides the recommended action, and $\mathbf{v}_e \in \mathbb{R}^d$ is the semantic embedding for retrieval. The combined text length satisfies $|c| + |a| \leq L_{\text{max}}^e$ words, bridging the gap between general workflows and specific environmental constraints.

With these definitions, we formalize the multimodal tool-use task as a Partially Observable Markov Decision Process (POMDP). This formulation is motivated by the fact that visual observations provide only partial information about the underlying task state: an image may reveal object appearance but not its identity or the broader context required for correct reasoning. Let $\mathcal{T} = (q, \mathcal{I})$ denote a task instance, where $q$ is a natural language query and $\mathcal{I} = \{I_1, I_2, \ldots, I_m\}$ is a set of relevant images. The agent is equipped with a toolset $\mathcal{F} = \{f_1, f_2, \ldots, f_n\}$, including capabilities such as code execution and web search. At each time step $t$, the agent receives an observation $o_t$ and generates an action $a_t \in \mathcal{F}$ based on the current state $s_t$, yielding a trajectory $\tau = [(s_0, a_0, o_0), \ldots, (s_T, a_T, o_T)]$ that concludes with a final answer $\hat{y}$. The objective is to construct an external knowledge base $\mathcal{KB} = (\mathcal{K}, \mathcal{E})$ that, when combined with $\text{MLLM}_{\text{exec}}$, maximizes the probability of generating the correct answer: $\max_{\mathcal{KB}} \mathbb{P}[\hat{y} = y^* \mid \mathcal{T}, \mathcal{KB}]$.

## 2.2. Phase I: Accumulation of Task Experience & Skills

### 2.2.1. ROLLOUT SUMMARY

For each training task $\mathcal{T}_i = (q_i, \mathcal{I}_i)$, $\text{MLLM}_{\text{exec}}$ performs $N$ independent rollouts to generate a trajectory set $\mathcal{R}_i = \{\tau_i^{(1)}, \ldots, \tau_i^{(N)}\}$. To handle long-context multimodal inputs, $\text{MLLM}_{\text{kb}}$ performs visually-grounded rollout summarization, taking the trajectory set $\mathcal{R}_i$, task images $\mathcal{I}_i$, query $q_i$, ground truth $y_i^*$, and the adapted skill $\mathcal{K}_{\text{adapted}}$ as input, producing both a trajectory summary and skill fragments:

$$\mathcal{S}_{\mathcal{R}_i}, \Delta\mathcal{K}_i = \text{MLLM}_{\text{kb}}(\mathcal{R}_i, \mathcal{I}_i, q_i, y_i^*, \mathcal{K}_{\text{adapted}}). \quad (1)$$

The summary $\mathcal{S}_{\mathcal{R}_i}$ contains key decision points, tool usage patterns, and failure reasons.

To bridge the visual-semantic gap, the summarization process grounds knowledge extraction in visual observations rather than relying only on textual trajectory logs. Concretely, $\text{MLLM}_{\text{kb}}$ receives interleaved image observations together with trajectory text and analyzes each image jointly with its local generation context, including tool calls, intermediate outputs, and task requirements. The resulting summary records not only what action was taken, but also what visual evidence motivated it and how that evidence affected subsequent decisions. For example, it may identify that an inverted image triggered rotation or that low contrast motivated image enhancement. These analyses are then integrated into $\mathcal{S}_{\mathcal{R}_i}$ to preserve the link between visual states and reasoning decisions. Simultaneously, skill extraction produces $\Delta\mathcal{K}_i$, which contains skill fragments abstracted from successful trajectories. Workflow extraction identifies structured sequences, while tool template extraction captures reusable code patterns. Specific entities are replaced with variable placeholders to ensure cross-task generalization. Prompt details and examples are provided in Appendix C.2.2 and Appendix C.2.4.

### 2.2.2. CROSS-ROLLOUT CRITIQUE

After the summarization of each rollout, a cross-rollout critique mechanism is employed to distill generalized knowledge from $\mathcal{R}_i$. Given the trajectory summary $\mathcal{S}_{\mathcal{R}_i}$, ground truth $y_i^*$, and the experiences $\mathcal{E}_{\text{ret}}$ that were used during the rollouts, $\text{MLLM}_{\text{kb}}$ performs contrastive analysis between successful and failed trajectories to identify causal factors behind outcomes:

$$\Delta\mathcal{E}_i = \text{MLLM}_{\text{kb}}(\mathcal{S}_{\mathcal{R}_i}, y_i^*, \mathcal{E}_{\text{ret}}). \quad (2)$$

The critique outputs structured experience updates $\Delta\mathcal{E}_i = \{op_1^e, op_2^e, \ldots, op_{M_i}^e\}$, where each operation takes the form $(\text{action}, \text{args})$ with $\text{action} \in \{\text{add}, \text{modify}\}$. The $(\text{add}, e)$ operation introduces a new experience $e \in \mathcal{E}$, while $(\text{modify}, e_{\text{id}}, e')$ refines an existing experience $e_{\text{id}}$ to $e'$. Each experience is constrained to $\leq L_{\text{max}}^e$ words and must

be generalizable across similar problem instances, enforced through prompt constraints in the generation process.

### 2.2.3. KNOWLEDGE CONSOLIDATION

To ensure the scalability and quality of the knowledge base, we implement a hierarchical consolidation mechanism that transforms transient trajectory insights into persistent, generalizable wisdom. The consolidation process handles both explicit operations from critique outputs and implicit operations triggered by similarity or quality thresholds.

For experience consolidation, before committing each $(\text{add}, e)$ operation, the system checks whether any existing entry has cosine similarity above $\theta_{\text{sim}}$ with $e$. If so, $e$ and all similar entries are jointly provided to $\text{MLLM}_{\text{kb}}$, which produces a single merged entry that preserves the core insights; the original entries are then removed. Otherwise, $e$ is added directly. The $(\text{modify}, e_{\text{id}}, e')$ operation updates the target entry in place. When the repository size exceeds $N_{\text{max}}^E$, the experience manager performs $(\text{delete}, o_{\text{id}})$ operations to remove redundant or low-quality items, where quality is assessed by $\text{MLLM}_{\text{kb}}$ based on generalizability and actionability.

For skill consolidation, fragments in $\Delta\mathcal{K}_i$ are integrated into the global skill document $\mathcal{K}$ by jointly providing both to $\text{MLLM}_{\text{kb}}$, which decides for each section whether to update, merge, or remove content. The skill manager monitors the document length and triggers refinement when it exceeds $L_{\text{max}}^K$. During refinement, $\text{MLLM}_{\text{kb}}$ assesses generalizability, workflow correctness, and conciseness of each component via self-evaluation, removing overly specific details and replacing concrete instances with reusable placeholders. Merge and refinement prompts are provided in Appendix C.2.2 and Appendix C.2.4.

## 2.3. Phase II: Solving Task with Experience & Skills

During inference, the agent faces a test task $\mathcal{T}_{\text{test}}$. Instead of relying on static prompting, we employ a dynamic retrieval and injection mechanism comprising three steps: task-decomposed retrieval, context-aware visual adaptation, and non-prescriptive injection.

### 2.3.1. TASK DECOMPOSITION RETRIEVAL

Directly retrieving experiences using the raw query $q$ often yields suboptimal results due to visual-semantic gaps and query specificity. In complex multimodal scenarios, the agent may simultaneously require experiences that address different technical aspects. To this end, we employ a task decomposition strategy: given the task description $q$ and images $\mathcal{I}$, $\text{MLLM}_{\text{kb}}$ decomposes the task into $n_g$ abstract subtasks $\mathcal{G} = \{g_1, \ldots, g_{n_g}\}$ that capture distinct needs such as dark image handling, geometric comparison, or error re-

covery. For each subtask, we generate a short textual query from the task description and visual context, and use it to retrieve relevant experiences. This multi-aspect decomposition followed by subtask-specific querying allows retrieval to target individual technical needs more precisely while maintaining broad coverage across problem dimensions.

For each subtask query, we compute its embedding $\mathbf{v}_g$ and retrieve the top-$k$ relevant experiences by comparing with their embeddings:

$$\mathcal{E}_{\text{ret}} = \bigcup_{g \in \mathcal{G}} \text{Top-k} \left( \{ e \in \mathcal{E} \mid \cos(\mathbf{v}_g, \mathbf{v}_e) > \tau_{\min} \} \right). \quad (3)$$

This decomposition-based retrieval significantly improves coverage compared to single-query retrieval, ensuring that critical experiences across multiple aspects are not overlooked and that retrieved guidance is better aligned with the actual technical demands of the task.

### 2.3.2. TASK ADAPTATION & INJECTION

**Experience Rewrite.** Retrieved experiences are generic and may contain irrelevant details for the current task. We therefore introduce an experience rewriter rather than injecting retrieved entries directly. Given the retrieved experiences, task description $q$, and task images $\mathcal{I}$, $\text{MLLM}_{\text{kb}}$ rewrites each experience while preserving its structure. It rephrases the condition to match the current task and visual state, instantiates the action with task-relevant details, and discards experiences that are clearly inapplicable. For example, a generic instruction to normalize image orientation may be rewritten as rotating an upside-down image before object detection. The output is a set of context-specific reformulations:

$$\mathcal{E}_{\text{rewritten}} = \text{MLLM}_{\text{kb}}(\mathcal{E}_{\text{ret}}, q, \mathcal{I}). \quad (4)$$

This keeps the guidance actionable and grounded in the current visual context while filtering out irrelevant entries; concrete prompt formulations are provided in Appendix C.2.5 and Appendix C.2.3.

**Skill Adaptation.** The global skill document $\mathcal{K}$ contains comprehensive workflows that may be overly detailed for the current task. We therefore adapt the retrieved skill to the current multimodal context rather than using it as a fixed template. The skill adaptor utilizes $\text{MLLM}_{\text{kb}}$ to prune irrelevant sections using the current images and task description, integrate the rewritten experiences into workflow steps, and adjust code templates to be task-relevant:

$$\mathcal{K}_{\text{adapted}} = \text{MLLM}_{\text{kb}}(\mathcal{K}, \mathcal{E}_{\text{rewritten}}, q, \mathcal{I}). \quad (5)$$

The adaptation process accesses task images $\mathcal{I}$ to perform visual-semantic alignment, ensuring that the adapted knowledge is grounded in the current visual context. The resulting adapted skill is then injected into the system prompt of the agent as a non-prescriptive reference, allowing the agent to leverage established wisdom while retaining the flexibility to improvise novel solutions when circumstances deviate from prior experience.

During task execution, the system records which skills and experiences were actually used, forming a usage history $\mathcal{H}_{\text{usage}} = (\mathcal{K}_{\text{adapted}}, \mathcal{E}_{\text{ret}})$. This history is fed back to the accumulation phase as a reference to improve rollout summary and cross-rollout critique, enabling continuous refinement of the knowledge base based on real-world usage patterns.

## 3. Experiments

### 3.1. Experimental Setup

**Datasets.** To verify the effectiveness of the proposed framework across diverse scenarios, we evaluate XSKILL on five benchmarks categorized into three distinct domains. The first domain represents visual agentic tool use and focuses on visual reasoning with multi-tool manipulation. For this, we utilize VisualToolBench (Guo et al., 2025a) and TIR-Bench (Li et al., 2025a), which require agents to process visual inputs and execute precise tools for image analysis. The second domain emphasizes multimodal search through information retrieval and web browsing. We employ MMSearch-Plus (Tao et al., 2025) and MMBrowseComp (Li et al., 2025c) to challenge the ability of agents to search and reason over heterogeneous textual and visual sources. Finally, we include AgentVista (Su et al., 2026), an ultra-challenging comprehensive benchmark that combines both tool use and complex search tasks. For each benchmark, we partition the data by randomly sampling 100 tasks to construct a training set for experience accumulation while reserving the remaining tasks for evaluation. Detailed statistics of the datasets are provided in Appendix B.1.

**Baselines.** We compare our method against the following state-of-the-art learning-from-experience baselines: (1) **Agent Workflow Memory (AWM)** (Wang et al., 2024), which extracts reusable task workflows from past trajectories to guide future task execution; (2) **Dynamic CheatSheet (DC)** (Suzgun et al., 2025), which maintains an evolving memory of problem-solving strategies and code snippets, enabling test-time learning without parameter updates; (3) **Agent-KB** (Tang et al., 2025), which aggregates crossdomain experiences into a structured knowledge base and employs hybrid retrieval to provide planning guidance and diagnostic feedback. For a fair comparison, all methods collect experience on the same training sets and perform experience retrieval at inference time, ensuring consistent experimental settings. More details regarding baseline implementations can be found in Appendix B.3.

**Tool Sets.** We equip the agent with a flexible library of tools

*Table 1.* Active tool sets for each dataset. All baselines and our method use these specific toolsets. **Search-W** represents web search, and **Search-I** represents image search.

| Dataset | Code | Search-W | Search-I | Visit |
|---|---|---|---|---|
| *Visual Agentic Tool Use* | | | | |
| VisualToolBench | ✓ | ✓ | – | ✓ |
| TIR-Bench | ✓ | – | – | – |
| *Multimodal Search* | | | | |
| MMSearch-Plus | ✓ | ✓ | ✓ | ✓ |
| MMBrowseComp | ✓ | ✓ | ✓ | ✓ |
| *Comprehensive* | | | | |
| AgentVista | ✓ | ✓ | ✓ | ✓ |

tailored to each task category (see Table 1). The specific functionalities are defined as follows: (1) code interpreter: executes Python code for image processing, calculations, and data analysis; (2) web search: retrieves relevant information and links from the web using text-based queries; (3) image search: performs reverse image searches to retrieve information and links associated with a given image query; (4) visit: browses specific webpages to extract content via a webpage link. Full definitions are provided in Appendix C.1.

**Metrics.** The primary evaluation metric is **Success Rate (SR)**. We conduct $N = 4$ rollouts for each task to evaluate the performance of the agent. Specifically, we report: (1) **Average@$N$** (with $N = 4$): the average success rate across all $N$ rollouts, which serves as a robust estimator of the reliability of the agent; (2) **Pass@$N$** (with $N = 4$): the proportion of tasks where at least one rollout is successful, reflecting the upper-bound capability of the agent.

**Implementation Details.** We evaluate our framework using Gemini-2.5-Pro (Comanici et al., 2025), Gemini-3-Flash (Google DeepMind, 2025), GPT-5-mini (OpenAI, 2025b), and o4-mini (OpenAI, 2025a). We also conducted further analysis of our framework on the open-source models Qwen3-VL-235B-Instruct and Qwen3-VL-32B-Instruct (Bai et al., 2025) (see Appendix A). In the main experiment, Gemini-2.5-Pro and Gemini-3-Flash accumulate experiences and skills from their own reasoning trajectories, while GPT-5-mini and o4-mini directly use knowledge accumulated by Gemini-3-Flash to examine cross-model transferability. For indexing, we use text-embedding-3-small (OpenAI, 2024). During inference, we set the retrieval top-$k$ to 3. Generation parameters are set to temperature $T = 0.6$, top-$p = 1.0$, and max turns = 20. Further details are listed in Appendix B.

### 3.2. Main Results

Table 2 presents the in-distribution performance comparison across four benchmarks and four backbone models.

Our method demonstrates substantial improvements over the tool-only baseline, achieving average gains of 2.58 to 6.71 points in Average@4 across different models. This validates that learning from accumulated experiences and skills provides substantial additional benefits on top of tool access. Compared to learning-based baselines, our method outperforms prior approaches in most settings, with particularly pronounced advantages on tasks requiring complex visual reasoning and multi-step tool composition. For instance, on TIR-Bench with Gemini-3-Flash, our method achieves 47.75% Average@4, surpassing the strongest baseline Agent-KB by 11.13 points. This demonstrates that the dual-stream design effectively captures knowledge that is helpful for solving the tasks. The improvements also extend to models using transferred knowledge: GPT-5-mini and o4-mini gain 2.58 to 4.16 points over the tool-only baseline, suggesting that externalized knowledge structures remain effective across different model architectures without requiring model-specific accumulation. Across all settings, the consistent improvements in both Average@4 and Pass@4 metrics indicate that our approach not only enhances the reliability of individual rollouts, but also improves the upper-bound capability.

### 3.3. Ablation Study

To analyze the contribution of individual components, we conduct a systematic ablation study on VisualToolBench using Gemini-2.5-Pro as shown in Table 3. Removing either experiences or skills leads to performance drops of 3.04 and 3.85 points respectively, validating that both knowledge streams are essential. When examining the two-phase architecture, we find that Phase 1 components (Experience Manager and Skill Manager) contribute more substantially than Phase 2 components (Task Decomposition and Task Adaptation), with drops of 4.09 and 3.62 points versus 1.28 and 1.52 points. This indicates that the quality of accumulated knowledge is more critical than the retrieval mechanism, though both phases remain necessary for optimal performance. These results confirm that each component addresses distinct challenges in multimodal agentic reasoning, and their integration is key to the effectiveness of the framework. To further understand how experiences and skills contribute to this effectiveness, we analyze the specific behavioral patterns induced by each knowledge stream in the following subsections.

**Skills Mitigate Inefficient Tool Use.** We first examine how skills contribute to tool-use efficiency by analyzing execution errors on VisualToolBench with Gemini-2.5-Pro. As shown in Figure 3, the transition from the Experience Only setting to the Skill Only setting results in a substantial decrease in execution failures. The overall error rate drops from 29.9% (168 errors) to 15.3% (95 errors), which shows that skills provide a strong foundation for reliable tool use.

*Table 2.* Main results of performance comparison (%) between XSKILL and baselines. We report the Average@4 and Pass@4 over 4 independent rollouts. **Bold** indicates the best performance.

| Model | Methods | VisualToolBench | | TIR-Bench | | MMSearch-Plus | | AgentVista | | Avg | |
|---|---|---|---|---|---|---|---|---|---|---|---|
| | | Average@4 | Pass@4 | Average@4 | Pass@4 | Average@4 | Pass@4 | Average@4 | Pass@4 | Average@4 | Pass@4 |
| Gemini-2.5-Pro | No Tools | 20.91 | 28.97 | 21.37 | 40.00 | 10.43 | 19.43 | 17.89 | 28.44 | 17.65 | 29.21 |
| | w/ Tools | 25.35 | 40.65 | 28.38 | 54.00 | 21.56 | 35.55 | 20.18 | 33.94 | 23.87 | 41.04 |
| | AWM | 25.93 | 39.25 | 29.75 | 53.50 | 20.85 | 36.97 | 19.72 | 32.11 | 24.06 | 40.46 |
| | DC | 24.77 | 37.38 | 27.62 | 51.00 | 24.64 | 40.76 | 21.79 | **35.78** | 24.71 | 41.23 |
| | Agent-KB | 26.75 | 41.12 | 29.13 | 52.50 | 23.22 | 37.91 | 20.87 | 33.94 | 24.99 | 41.37 |
| | **XSkill** | **30.49** | **46.73** | **33.12** | **58.00** | **27.96** | **44.08** | **22.94** | 34.86 | **28.63** | **45.92** |
| Gemini-3-Flash | No Tools | 25.12 | 36.92 | 28.50 | 54.00 | 16.47 | 24.64 | 18.35 | 29.36 | 22.11 | 36.23 |
| | w/ Tools | 41.94 | 60.75 | 32.37 | 58.50 | 39.57 | 53.55 | 20.64 | 39.45 | 33.63 | 53.06 |
| | AWM | 41.94 | 59.35 | 34.25 | 62.50 | 43.36 | 54.98 | 19.50 | 37.61 | 34.76 | 53.61 |
| | DC | 41.70 | 59.81 | 33.75 | 59.00 | 40.28 | 55.45 | 20.18 | 36.70 | 33.98 | 52.74 |
| | Agent-KB | 41.75 | 61.21 | 36.62 | 62.00 | 39.81 | 53.08 | 21.33 | 38.53 | 34.88 | 53.71 |
| | **XSkill** | **46.50** | **64.02** | **47.75** | **75.00** | **43.72** | **56.40** | **23.39** | **40.37** | **40.34** | **58.95** |
| GPT-5-mini | No Tools | 13.90 | 22.90 | 20.00 | 46.50 | 3.08 | 6.64 | 18.58 | 28.44 | 13.89 | 26.12 |
| | w/ Tools | 24.30 | 37.85 | 23.50 | 50.50 | 14.22 | 20.38 | 20.41 | 35.78 | 20.61 | 36.13 |
| | AWM | 23.25 | 34.58 | 24.25 | 53.00 | 14.81 | 20.85 | 19.04 | 34.86 | 20.34 | 35.82 |
| | DC | 23.83 | 35.05 | 24.50 | 53.50 | 13.74 | 18.48 | 21.10 | 36.70 | 20.79 | 35.93 |
| | Agent-KB | **24.77** | **38.32** | 25.13 | 53.00 | 13.63 | 19.91 | 19.95 | 34.86 | 20.87 | 36.52 |
| | **XSkill** | 24.53 | 37.85 | **28.25** | **56.00** | **16.11** | **23.22** | **23.85** | **38.53** | **23.19** | **38.90** |
| o4-mini | No Tools | 14.72 | 27.57 | 20.13 | 45.00 | 5.92 | 12.32 | 19.04 | 25.69 | 14.95 | 27.65 |
| | w/ Tools | 19.63 | 34.11 | 24.62 | 49.50 | 15.88 | 21.80 | 18.12 | 29.36 | 19.56 | 33.69 |
| | AWM | 21.14 | 36.92 | 25.25 | 51.00 | 16.94 | 22.75 | 19.95 | 29.36 | 20.82 | 35.01 |
| | DC | 20.44 | 33.64 | 25.50 | 52.00 | 14.57 | 22.27 | 18.81 | 28.44 | 19.83 | 34.09 |
| | Agent-KB | 22.78 | 36.92 | 24.75 | 52.50 | 15.17 | 20.38 | 19.50 | 31.19 | 20.55 | 35.25 |
| | **XSkill** | **25.00** | **41.12** | **30.25** | **57.50** | **17.30** | **23.70** | **22.32** | **33.94** | **23.72** | **39.07** |

*Table 3.* Ablation study of performance (%) on VisualToolBench using Gemini-2.5-Pro. We systematically remove key components to evaluate their contribution. $\Delta$ represents the absolute performance drop compared to the full pipeline.

| Setting | Average@4 | Pass@4 | $\Delta$ Avg@4 |
|---|---|---|---|
| **Ours - Full Pipeline** | **30.49** | **46.73** | – |
| *Experience & Skill Ablation* | | | |
| w/o Experience | 27.45 | 42.52 | -3.04 |
| w/o Skill | 26.64 | 41.12 | -3.85 |
| *Phase 1 Ablation* | | | |
| w/o Experience Manager | 26.40 | 42.06 | -4.09 |
| w/o Skill Manager | 26.87 | 42.99 | -3.62 |
| *Phase 2 Ablation* | | | |
| w/o Task Decomposition | 29.21 | 44.86 | -1.28 |
| w/o Task Adaptation | 28.97 | 44.39 | -1.52 |
| w/ Tools | 25.35 | 40.65 | -5.14 |
| No Tools | 20.91 | 28.97 | -9.58 |

This directly addresses the problem of inefficient tool use mentioned in Section 1. A detailed breakdown shows that skills effectively reduce structural mistakes: syntax errors decrease from 114 (20.3%) to 71 (11.4%), and tool name errors are almost entirely removed (16 to 2). By providing clear tool templates and workflow instructions, skills prevent the agent from wasting reasoning steps on error recovery.

This ensures that more of the limited turn budget is used for productive problem solving.

**Experiences Enable Flexible Orchestration.** We next evaluate how experiences influence tool selection by analyzing the distribution of tool usage (Table 4). The introduction of experiences leads to a clear shift in tool use patterns that skills alone do not produce. On VisualToolBench, the Experience Only setting increases code interpreter usage from 66.63% to 74.49%, a trend that continues in the full pipeline (76.97%). Similarly, on MMSearch-Plus, experiences double the use of the code interpreter (6.18% to 13.21%) and increase image search calls (15.43% to 24.63%). These changes show that experiences capture tactical knowledge for specific tasks: for visual reasoning, they favor code-based processing; for multimodal search, they prioritize specialized visual tools over general text search. This context-aware adaptation allows XSKILL to move beyond the fixed execution paths of baselines, achieving the flexible tool orchestration needed for complex multimodal tasks.

## 3.4. Analysis

**Impact of Rollout Values.** To investigate how the number of rollouts affects performance, we vary $N \in \{1, 2, 3, 4\}$ on VisualToolBench with Gemini-2.5-Pro and o4-mini. Figure 4 shows that both Average@4 and Pass@4 improve

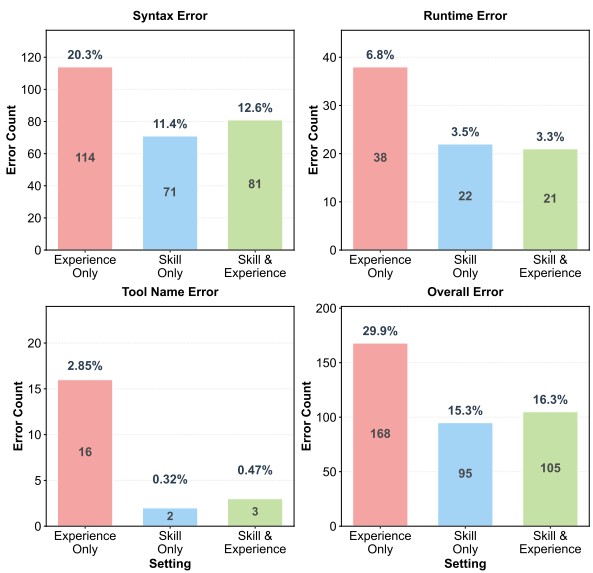

*Figure 3.* Error analysis on VisualToolBench using Gemini-2.5-Pro. Error counts (inside bars) and their proportions relative to total tool calls (above bars) are compared across three settings. Skills significantly reduce syntax and runtime errors, leading to more robust tool execution.

*Table 4.* Tool usage distribution (%) on VisualToolBench and MMSearch-Plus using Gemini-2.5-Pro. Experiences shift tool selection towards more targeted strategies.

| Setting | Code | Search-W | Search-I | Visit |
|---|---|---|---|---|
| *VisualToolBench* | | | | |
| w/ Tools | 66.63 | 31.70 | – | 1.66 |
| Skill Only | 65.96 | 31.10 | – | 2.82 |
| Exp Only | **74.49** (↑) | **22.03** (↓) | – | 2.56 |
| **Skill & Exp** | **76.97** (↑) | **21.12** (↓) | – | 1.58 |
| *MMSearch-Plus* | | | | |
| w/ Tools | 6.18 | 71.07 | 15.43 | 7.32 |
| Skill Only | 7.94 | 67.07 | 17.87 | 7.12 |
| Exp Only | **13.21** (↑) | **56.12** (↓) | **24.63** (↑) | 6.04 |
| **Skill & Exp** | **14.37** (↑) | **55.08** (↓) | **23.89** (↑) | 5.66 |

consistently as $N$ increases, with Pass@4 exhibiting steeper gains. This scaling behavior stems from the accumulation phase: more rollouts provide richer trajectory diversity, enabling the cross-rollout critique mechanism to extract higher-quality experiences by contrasting successful and failed attempts, and to induce more generalizable skills by identifying common patterns across varied execution paths. During inference, this improved knowledge base allows agents to recognize promising solution strategies more effectively, leading to better performance across multiple independent trials.

**Cross-Task Transferability.** To evaluate the generalization capability of accumulated knowledge, we conduct zero-shot transfer experiments where knowledge from one benchmark is applied to a different benchmark without target-domain training. Specifically, we use VisualToolBench knowledge

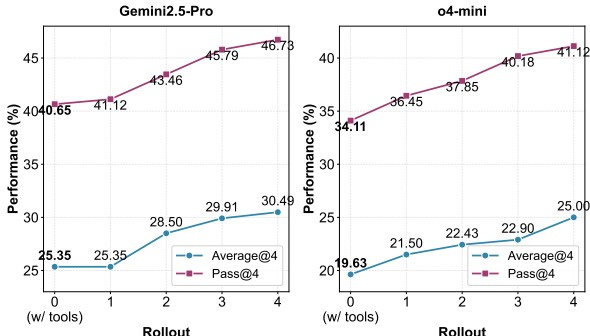

*Figure 4.* Performance comparison across different rollout values on VisualToolBench. Rollout $N = 0$ corresponds to the baseline with tools (w/ tools). The results show consistent improvement as the number of rollouts increases.

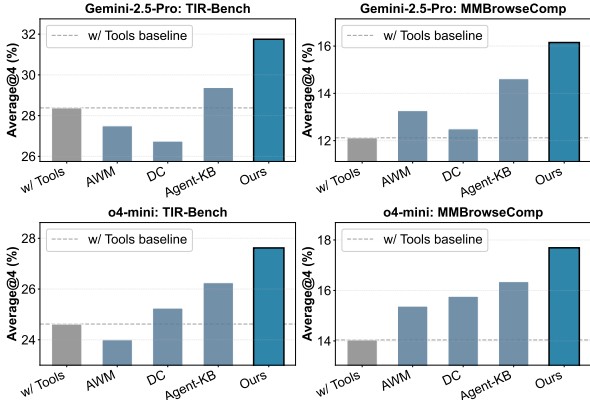

*Figure 5.* Out-of-distribution performance comparison (Average@4) of different methods on TIR-Bench and MMBrowseComp. The gray horizontal dashed line represents the w/ Tools baseline. Our method (highlighted with black border) consistently outperforms all baseline methods across both models and benchmarks.

to solve TIR-Bench tasks, and MMSearch-Plus knowledge for MMBrowseComp tasks. Figure 5 shows that our method consistently outperforms all baselines across both target benchmarks and backbone models, with average improvements of 2 to 3 points over Agent-KB. The superior transferability stems from the hierarchical consolidation mechanism that removes case-specific details while preserving broadly applicable insights, and the task-adaptation process, which tailors experiences and skills to fit the current task context.

## 4. Related Work

### 4.1. Multimodal Agentic Reasoning

The evolution of MLLMs has shifted the paradigm from static visual understanding to active "thinking with images" (Su et al., 2025). Beyond passive perception, modern agents leverage diverse toolsets to manipulate and analyze visual data: they can actively zoom or adjust image properties to clarify details (Zheng et al., 2025b; Wang et al., 2025a), synthesize executable code for precise visual trans-

formations (Guo et al., 2025c; Zhao et al., 2025), and orchestrate web searches to retrieve context based on visual cues (Geng et al., 2025; Chu et al., 2026). Despite these capabilities, the majority of existing agentic frameworks remain fundamentally stateless. As noted in recent literature (Li et al., 2025d; Liu et al., 2025), current multimodal systems operate in isolated episodes, which prevents the internalization of successful tool-use patterns or corrective feedback across tasks. This limitation results in redundant trial-and-error, necessitating a shift toward evolving agents that can accumulate cross-episode procedural expertise from lifelong interactions. While parametric approaches such as reinforcement learning have been proposed to internalize such strategies (Hong et al., 2025; Geng et al., 2025; Chu et al., 2026), they face significant scalability bottlenecks due to the high cost of domain-specific training and the difficulty of adapting to evolving toolsets. Consequently, there is a pressing need for non-parametric mechanisms that allow agents to accumulate reasoning capabilities continuously and flexibly.

### 4.2. Learning from Experience and Skills

Equipping agents with the capability to learn from past interactions has emerged as a compelling alternative to the prohibitive costs of continuous parameter fine-tuning. Early approaches empowered agents by retrieving raw execution trajectories to inform decision-making (Zheng et al., 2023; Zhao et al., 2024). To enhance generalization, recent research has shifted toward abstracting these raw traces into reusable knowledge, which typically manifests in two complementary forms: ❶ **Experiences**, which capture tactical, condition-action insights for specific contexts (Tang et al., 2025; Cai et al., 2025); and ❷ **Skills**, which encode high-level procedural workflows and reusable templates (Wang et al., 2024; Zheng et al., 2025a; Anthropic, 2026; Wang et al., 2025b). Furthermore, to facilitate autonomous self-improvement, frameworks such as EvolveR (Wu et al., 2025a) and ReasoningBank (Ouyang et al., 2025) introduce closed-loop evolutionary lifecycles to refine and consolidate this knowledge. Despite these successes in textual domains, experience learning in multimodal settings remains significantly underexplored. Existing multimodal memory attempts (Li et al., 2025b; Wu et al., 2025b) are often confined to specialized tasks such as GUI navigation or spatial reasoning. More importantly, when retrieving prior knowledge, they typically rely on the raw textual instruction, without plan-then-retrieve mechanisms grounded in the visual context of the new problem. They also do not adapt retrieved experiences or tool templates to the current multimodal context. Our work addresses these limitations through a unified framework for visually grounded knowledge extraction and context-aware adaptation.

## 5. Conclusion

Our work addresses a fundamental limitation in multimodal agents: the lack of effective mechanisms for leveraging knowledge from past interactions. While existing approaches rely on static documentation or text-only extraction, we show that effective knowledge accumulation in multimodal settings requires grounding insights in visual observations for both task-level and action-level knowledge. To this end, we propose XSKILL, a framework that unifies task-level skills with action-level experiences through visually grounded extraction and hierarchical consolidation. Visually grounded retrieval and adaptation further bridge the gap between accumulated knowledge and task-specific requirements during inference. Empirical validation across diverse benchmarks confirms the effectiveness of XSKILL, yielding consistent performance improvements over strong baselines. The results of ablation studies reveal that these dual knowledge streams provide distinct yet complementary advantages: skills ensure the robustness of tool execution, while experiences guide strategic selection based on task-specific contexts. Furthermore, the strong zero-shot transferability of the framework suggests that it captures generalizable reasoning principles rather than simple heuristics. While our current evaluation demonstrates a single accumulation-then-test cycle, the architecture of the framework is compatible with iterative refinement as new tasks are encountered. By enabling multimodal agents to accumulate and leverage knowledge without the need for parametric retraining, XSKILL offers a practical and scalable path toward evolving autonomous systems in real-world environments.

## Acknowledgment

This work is supported in part by Grant Web26EG02. This work is also supported in part by the Open Research Fund of the Guangdong Laboratory of Artificial Intelligence and Digital Economy (SZ) (Grant No. GML-KF-26-06). We thank all collaborators for their helpful discussions and feedback throughout the development of this work. We are also grateful to the developers and maintainers of the open-source models and benchmarks used in our evaluation, whose contributions made this research possible.

## Impact Statement

This work introduces XSKILL, a framework that enables multimodal agents to accumulate and leverage task-level skills and action-level experiences for continual improvement without parameter updates. By externalizing knowledge into structured, human-readable representations, XSKILL improves the transparency and interpretability of agent decision-making, and the explicit separation of skills and experiences makes it possible for human operators to

audit, edit, or remove specific pieces of accumulated knowledge. However, more capable agents could be misused for malicious automation involving sensitive visual data, or may inherit and amplify biases present in previous trajectories through the accumulation loop. The cross-model transferability demonstrated in our experiments further implies that biased knowledge accumulated by one model could propagate to others without additional safeguards. To mitigate these concerns, we recommend human oversight for reviewing accumulated knowledge bases, periodic bias auditing of both skill documents and experience banks, and access control policies governing cross-model knowledge transfer.

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

# A. Additional Experimental Results

## A.1. Open-Source Model Evaluation

To assess the generalizability of the proposed framework to open-source models, we evaluate Qwen3-VL-235B-Instruct and Qwen3-VL-32B-Instruct on VisualToolBench and MMSearch-Plus. These models utilize knowledge accumulated by Gemini-3-Flash to examine cross-model transferability without requiring model-specific accumulation.

**Analysis.** Knowledge transfer from Gemini-3-Flash to open-source models shows mixed effectiveness. While the proposed method achieves gains on MMSearch-Plus for both models, it exhibits negative effects on the Average@4 performance on VisualToolBench compared to the tool-only baseline. This suggests that externally accumulated knowledge can interfere with the native tool-use behaviors of weaker models. However, XSKILL significantly increases exploratory behavior by encouraging more tool invocations, which translates to improvements in Pass@4 despite the degradation in Average@4. This pattern indicates that increased exploration provides more opportunities to reach correct answers across multiple rollouts, even when the average quality of individual trials is lower. These findings underscore that sufficient capabilities of the base model are critical for effective knowledge transfer.

*Table 5.* Performance comparison on open-source models (%). Knowledge is accumulated by Gemini-3-Flash and transferred to Qwen models. **Bold** indicates best. Avg Turns: average tool invocations per task.

| Model | Setting | VisualToolBench | | | MMSearch-Plus | | |
|---|---|---|---|---|---|---|---|
| | | Average@4 | Pass@4 | Turns | Average@4 | Pass@4 | Turns |
| **Qwen3-VL-235B** | No Tools | 10.51 | 17.76 | - | 2.49 | 6.16 | - |
| | w/ Tools | 11.80 | 19.62 | 3.82 | 8.76 | 14.69 | 3.83 |
| | Agent Workflow Memory | 10.04 | 18.22 | 3.71 | **12.32** | 20.38 | 4.03 |
| | Dynamic CheatSheet | 10.86 | 19.16 | 4.09 | 10.82 | 18.01 | 3.87 |
| | Agent-KB | **12.85** | 20.09 | 3.88 | 10.29 | 17.54 | 4.71 |
| | **XSkill** | 11.52 | **20.56** | 5.12 | 10.43 | **21.80** | 4.15 |
| **Qwen3-VL-32B** | No Tools | 7.94 | 12.62 | - | 2.37 | 4.74 | - |
| | w/ Tools | **11.09** | 18.69 | 2.65 | 9.95 | 19.91 | 2.88 |
| | Agent Workflow Memory | 10.05 | 17.76 | 2.73 | 9.48 | 18.48 | 2.77 |
| | Dynamic CheatSheet | 9.11 | 15.89 | 2.54 | 11.14 | 21.80 | 3.13 |
| | Agent-KB | 10.28 | 16.82 | 2.88 | 11.37 | 22.27 | 3.04 |
| | **XSkill** | 10.28 | **19.43** | 4.19 | **13.10** | **24.17** | 4.18 |

# B. Experimental Details

## B.1. Dataset Details

We evaluate XSKILL on five benchmarks spanning three distinct domains: visual agentic tool use, multimodal search, and comprehensive evaluation. For each dataset, we partition the data into training and test sets. The training set is used for accumulating experiences and skills through Phase I (accumulation), while the test set is used to evaluate the performance of the agent in Phase II (inference). All training and test sets are completely disjoint.

Table 6 summarizes the overall statistics and partitioning strategy for each benchmark. Note that MMBrowseComp is used exclusively for testing due to its limited size (130 samples), serving as an out-of-distribution evaluation target when transferring knowledge from MMSearch-Plus.

**VisualToolBench** (Guo et al., 2025a) focuses on hybrid tool reasoning tasks that require agents to process visual inputs and execute precise tools for image analysis. We specifically select single-turn tasks from the `hybrid_tool_reasoning` category to ensure consistent task complexity. From this subset, we randomly sample 100 tasks for training and 214 tasks for testing.

**TIR-Bench** (Li et al., 2025a) evaluates tool-integrated reasoning capabilities across diverse visual reasoning scenarios. The original dataset contains 1,215 samples across 13 categories. We filter the dataset to retain only 5 categories that are compatible with our tool set: `refcoco` (referring expression comprehension), `maze` (spatial reasoning), `instrument` (object detection), `ocr` (text recognition), and `contrast` (visual contrast analysis). This filtering yields 430 samples, from which we randomly sample 200 for testing and 100 for training. The category distributions are shown in Table 7.

*Table 6.* Dataset statistics and train/test split information. All datasets are partitioned with random seed 42 to ensure reproducibility.

| Dataset | Domain | Total | Train | Test | Partition Strategy |
|---|---|---|---|---|---|
| *Visual Agentic Tool Use* | | | | | |
| VisualToolBench | Hybrid Tool Reasoning | 1,191 | 100 | 214 | Single-turn hybrid tasks only |
| TIR-Bench | Tool-Integrated Reasoning | 1,215 | 100 | 200 | Filtered 5 categories (430 samples) |
| *Multimodal Search* | | | | | |
| MMSearch-Plus | Multimodal Search | 311 | 100 | 211 | Random sampling |
| MMBrowseComp | Multimodal Browsing | 130 | - | 130 | All samples for OOD testing |
| *Comprehensive* | | | | | |
| AgentVista | Ultra-challenging Tasks | 209 | 100 | 109 | Random sampling |

*Table 7.* Category distribution in TIR-Bench train and test splits. The distributions reflect the original proportions in the filtered dataset.

| Category | Train (100) | Test (200) |
|---|---|---|
| refcoco | 27 | 60 |
| maze | 27 | 51 |
| instrument | 22 | 39 |
| ocr | 15 | 28 |
| contrast | 9 | 22 |

**MMSearch-Plus** (Tao et al., 2025) challenges agents to search and reason over heterogeneous textual and visual sources through web and image search. The dataset contains 311 tasks, from which we randomly sample 100 for training and reserve the remaining 211 for testing.

**MMBrowseComp** (Li et al., 2025c) evaluates multimodal browsing and comprehension capabilities across web content. Due to its limited size (130 samples), we use all samples exclusively for testing, making it suitable for evaluating cross-task transferability when using knowledge accumulated from MMSearch-Plus.

**AgentVista** (Su et al., 2026) provides an ultra-challenging holistic evaluation combining both tool use and complex search tasks in realistic visual scenarios. The dataset contains 209 tasks, from which we randomly sample 100 for training and use the remaining 109 for testing.

### B.2. Hyperparameters

We provide the detailed hyperparameters used across all experiments in Table 8. The configurations are organized following the two-phase structure described in Section 2: Phase I focuses on accumulation of knowledge from training trajectories, while Phase II handles inference on test tasks.

**Phase I Configuration.** During the accumulation phase, $\text{MLLM}_{\text{kb}}$ processes trajectories with a temperature of 0.6 to balance creativity and consistency in knowledge extraction. Rollout summarization uses up to 12,288 tokens to accommodate long multimodal trajectories. The cross-rollout critique extracts up to 4 experience operations per task, with each experience limited to $L_{\max}^e = 64$ words. During consolidation, experiences with cosine similarity above $\theta_{\text{sim}} = 0.70$ are merged to reduce redundancy. When the experience library exceeds $L_{\max}^{\mathcal{E}} = 120$ items, the system performs aggressive pruning. Skills are refined when exceeding $L_{\max}^K = 1000$ words.

**Phase II Configuration.** For inference, task decomposition employs a lower temperature (0.3) with 2,048 tokens to ensure focused subtask generation. Each subtask retrieves the top-$k = 3$ most similar experiences using OpenAI's `text-embedding-3-small` model with caching enabled. Experience rewriting and skill adaptation uses temperature 0.3 and 8,192 tokens to adapt generic advice to task-specific contexts.

### B.3. Baseline Details

We implement three learning-from-experience baselines following their default configurations as specified in the original papers. To ensure fair comparison, all baselines are adapted to use the same training/test splits, backbone models ($\text{MLLM}_{\text{exec}}$ and $\text{MLLM}_{\text{kb}}$), and inference settings as XSKILL. Table 9 summarizes the key configurations that remain aligned with each

*Table 8.* Hyperparameters used in experiments, organized by the two-phase framework. MLLM$_{exec}$ denotes the execution model for task inference, and MLLM$_{kb}$ denotes the knowledge base model for all learning and adaptation operations.

| Parameter | Value | Description |
|---|---|---|
| *General Inference (MLLM$_{exec}$)* | | |
| Temperature ($T$) | 0.6 | Sampling temperature |
| Top-$p$ | 1.0 | Nucleus sampling parameter |
| Max Tokens per Turn | 8192 | Maximum completion tokens per turn |
| Max Turns | 20 | Maximum interaction turns per task |
| Max Images | 100 | Maximum images per task |
| Rollouts per Sample ($N$) | 4 | Number of independent rollouts |
| *Phase I: Rollout Summary (MLLM$_{kb}$)* | | |
| Summarization Temperature | 0.6 | Temperature for trajectory summarization |
| Summarization Max Tokens | 12288 | Max tokens for rollout summary |
| Image Summary Max Tokens | 2048 | Max tokens for image description |
| *Phase I: Cross-Rollout Critique (MLLM$_{kb}$)* | | |
| Critique Temperature | 0.6 | Temperature for experience extraction |
| Critique Max Tokens | 12288 | Max tokens for critique generation |
| Max Experience Length ($L^e_{\max}$) | 64 | Max words per experience item |
| Max Operations per Sample | 4 | Max experience operations per task |
| *Phase I: Knowledge Consolidation (MLLM$_{kb}$)* | | |
| Merge Threshold ($\theta_{\text{sim}}$) | 0.70 | Similarity threshold for merging |
| Max Experience Items ($L^{\mathcal{E}}_{\max}$) | 120 | Maximum experiences in library |
| Max Skill Document Length ($L^K_{\max}$) | 1000 | Word count threshold for skill refinement |
| Embedding Model | text-embedding-3-small | OpenAI embedding model |
| *Phase II: Task Decomposition Retrieval (MLLM$_{kb}$)* | | |
| Decomposition Temperature | 0.3 | Temperature for task decomposition |
| Decomposition Max Tokens | 2048 | Max tokens for subtask generation |
| Retrieval Top-$k$ | 3 | Number of experiences per subtask |
| Min Similarity ($\tau_{\min}$) | 0.0 | Minimum cosine similarity threshold |
| *Phase II: Task Adaptation & Injection (MLLM$_{kb}$)* | | |
| Experience Rewrite Temperature | 0.3 | Temperature for experience adaptation |
| Experience Rewrite Max Tokens | 8192 | Max tokens for experience rewriting |
| Skill Adaptation Temperature | 0.3 | Temperature for skill adaptation |
| Skill Adaptation Max Tokens | 8192 | Max tokens for skill adaptation |

method's default settings.

**Agent Workflow Memory (AWM)** (Wang et al., 2024) extracts reusable task workflows from past trajectories using LLM-based induction. The system abstracts specific execution details (e.g., filenames, URLs) into generic variable names while preserving logical steps and tool names. It retrieves only the single most similar workflow ($k = 1$) to provide focused guidance. Each workflow is constrained to 200 words to ensure conciseness. Following the default setting, AWM only accumulates knowledge from successful trajectories.

**Dynamic CheatSheet (DC)** (Suzgun et al., 2025) maintains an evolving memory of problem-solving strategies. For each query, it retrieves the most similar past trajectory ($k = 1$) and synthesizes a new global cheatsheet by combining it with the previous cheatsheet using an LLM. The cheatsheet is strictly limited to 200 words and updates dynamically at each query. Unlike other baselines, DC does not filter trajectories by success status, accumulating insights from all task executions as per its default design.

**Agent-KB** (Tang et al., 2025) aggregates cross-domain experiences into a structured knowledge base with four components: agent planning, search agent planning, agent experience, and search agent experience. It employs hybrid retrieval that combines TF-IDF-based text matching with semantic similarity (weighted 0.5 each) to retrieve $k = 3$ workflow instances. The system provides dual guidance: student guidance offering 2-3 planning suggestions based on similar task patterns, and teacher guidance providing unified operational instructions from experience entries. Query reasoning is enabled to optimize retrieval by extracting core concepts from raw queries using an LLM. Following the default setting, Agent-KB only accumulates knowledge from successful trajectories, with each knowledge component required to exceed 50 characters.

*Table 9.* Configuration parameters for baseline methods aligned with their default settings. All methods use text-embedding-3-small for retrieval and temperature 0.6 for knowledge extraction with MLLM$_{kb}$.

| Parameter | AWM | DC | Agent-KB |
|---|---|---|---|
| *Core Configuration* | | | |
| Retrieval Top-$k$ | 1 | 1 | 3 |
| Knowledge Format | Workflow | Cheatsheet | Dual Guidance |
| Max Knowledge Length | 200 words | 200 words | – |
| *Retrieval & Adaptation* | | | |
| Search Method | Semantic | Semantic | Hybrid (0.5/0.5) |
| Dynamic Update | No | Yes | No |
| Abstraction Method | LLM Induction | Synthesis | Query Reasoning |
| Success Filtering | Yes | No | Yes |

**Shared Settings Across Baselines.** To ensure consistent evaluation, we standardize several settings across all methods while preserving their core mechanisms. All baselines use text-embedding-3-small for semantic embedding and retrieval,. Trajectory summarization and knowledge extraction operations are performed using the same MLLM$_{kb}$ model as XSKILL with temperature 0.6. All methods use identical data splits as specified in Table 6, with the same 100 training samples per dataset for knowledge accumulation.

## C. Method Details

### C.1. Tool Definitions

We provide agents with four primary tools for multimodal task solving. Table 10 summarizes functionalities and parameters.

*Table 10.* Tool definitions and parameters used in experiments. All tools are accessible to agents via function calling.

| Tool | Description | Parameters |
|---|---|---|
| Web Search | Search the web for information online. Returns web search results with titles, URLs, and text snippets. | `query` (str, required): Search query string. `max_results` (int, default: 10): Maximum number of results. |
| Image Search | Search for related images using text query or reverse image search. Supports two modes: (1) text search with `search_type='text'`; (2) reverse image search with `search_type='reverse'`. | `search_type` (str, default: 'text'): Either 'text' or 'reverse'. `query` (str): Search query for text mode. `image_url` (str): Image reference for reverse mode. Supports 'original_image', 'tool_image_N', or image URLs. `max_results` (int, default: 10): Maximum results. |
| Visit | Visit a webpage and extract its main textual content. Typically used after obtaining URLs from search results. | `url` (str, required): Full URL starting with 'http://' or 'https://'. `goal` (str): Information to find on the page (helps focus extraction). |
| Code Interpreter | Executes Python code in a stateful Jupyter kernel. Supports image processing (PIL, OpenCV), calculations, and data manipulation. Pre-loaded image variables: `original_image`, `tool_image_N`. Pre-installed packages: PIL, NumPy, OpenCV, Matplotlib, SciPy, Scikit-learn, Pandas, SymPy. | `code` (str, required): Python code to execute. *Note:* Code execution is persistent across calls. Use `plt.show()` or save images to display outputs. |

**Image Reference Convention.** In Image Search and Code Interpreter, agents can reference images using the following naming convention: `original_image` refers to the initial input image(s), and `tool_image_N` refers to images generated by previous code executions (indexed by order, e.g., `tool_image_1`, `tool_image_2`). The Code Interpreter maintains a persistent execution state, allowing variables and functions defined in earlier calls to be reused in subsequent executions.

## C.2. Prompt Design

Detailed prompt design for each phase of the framework provided here.

### C.2.1. SYSTEM PROMPTS

---

**System Prompt: direct_cot**

You are a visual reasoning agent. Your goal is to answer questions about images.

INSTRUCTIONS:
1. Analyze: Carefully observe the image and the user's question.
2. Think: Explain your step-by-step reasoning process.
3. Answer: Once you are confident in your findings, you MUST provide the final answer inside `<answer> ...(your final answer)... </answer>` tags!
  E.g., `<answer>The answer is 10.</answer>`/`<answer> A </answer>`/...

---

**System Prompt: multi_tool_agent_search**

You are a visual reasoning agent. Your goal is to answer questions about images.

# AVAILABLE TOOLS:
You have access to the following tools:
1. **web_search**: Search the web for information, facts, or current events
2. **image_search**: Search for related images using text query or reverse image search
3. **visit**: Visit a webpage and extract its main content
4. **code_interpreter**: Execute Python code for image processing, analysis, and calculations

# INSTRUCTIONS:
1. Analyze: Carefully observe the image and the user's question.
2. Think: Explain your step-by-step reasoning process.
3. Use Tools: Call appropriate tool to gather information and help answer the question.
4. Iterate as needed: Continue reasoning and using tools in next turns until you are confident in your findings.
5. Answer: Once confident, provide the final answer inside `<answer>...</answer>` tags!

# IMPORTANT:
- Always explain your detailed reasoning process before using any tool.
- You can ONLY call one tool at one turn! Do not call multiple tools in one turn!
- You MUST provide your final answer using complete `<answer>...</answer>` tags!

---

### C.2.2. SKILL GENERATION PROMPTS

---

**GENERATE_RAW_SKILL_PROMPT**

You are a skilled AI agent architect. Analyze the trajectory and extract a reusable Standard Operating Procedure (SOP).
### Guiding Principles:
1. From successful patterns: Extract the effective workflows and tool sequences.
From failed attempts or near-misses: Note what went wrong and why - these lessons are often more valuable.
2. **Keep It General**: Use placeholders like `[TARGET]`, `[QUERY]` instead of specific values. The skill should apply to similar problems, not just this one.
3. **Capture Executable Knowledge**: When the trajectory includes effective code, extract the core logic as a reusable template. Good code templates are worth more than paragraphs of description.
4. **Brevity Matters**: Aim for ∼600 words. Focus on what's actionable.
### Output Structure:

---

---

```
name: [SkillName]
description: |
  [Clear, concise description of what this skill does and when to use it.
  1-2 sentences focusing on the core purpose and benefits.]
version: 1.0.0
---

# [Skill Title]

## Strategy Overview
[1-2 sentences on the core approach]

## Workflow
1. **[Phase Name]**: [Action and rationale]
2. **[Phase Name]**: [Action and rationale]
3. ...

## Tool Templates
(Include only if the trajectory contained useful code or query patterns)

- **[Tool] - [Purpose]**:
  ```python
  # [Brief comment]
  [code with placeholders]
  ```

- **Query Pattern**: ``[pattern with placeholders]''

## Watch Out For
- [Common mistake or trap from the trajectory]
```
### Input:
`<trajectory>  {trajectory}</trajectory>`

`<ground_truth>  {ground_truth}</ground_truth>`

Output ONLY the SKILL.md content starting with `---`.

---

## MERGE_SKILL_PROMPT

You are a knowledge architect. Your job is to maintain a single, unified skill document that grows wiser with each new case.
### Philosophy:
Think of the global skill as a living document. Each new skill brings potential insights - your task is to integrate them thoughtfully, not mechanically.

### Integration Strategy:
For each part of the new skill, ask:
- **Is this part better?** → Rewrite the existing version
- **Is this part redundant or too specific?** → Delete it
- **Is this part complementary?** → Merge into a more general form
- **Is this part genuinely different?** → Add as a variant workflow (but consolidate if possible)

### Quality Guidelines:
- Preserve concrete, reusable code templates and tool patterns
- Delete overly specific examples and cases that don't apply to similar problems
- Consolidate similar trigger phrases
- If workflows differ only in minor details, merge them into one with noted variations

### Length Budget:
- Target: ∼1000 words
- If growing too long: merge similar workflows, trim verbose explanations
- Maximum 4 variant workflows - if you have more, they likely can be consolidated

```
### Input:
<existing_skill>
  {existing_skill}
</existing_skill>

<new_skills>
  {new_skills}
</new_skills>
```

Output ONLY the merged SKILL.md starting with `---`. No preamble.

---

## SKILL_MANAGE_PROMPT

You are a skill document architect. Refine the SKILL.md to remove redundancy, generalize specific cases, and improve structure.
### Current Stats:
- Word count: {word_count}

### Refinement Goals:

1. **Remove Redundancy**:
   - Merge duplicate or near-duplicate content across sections
   - Eliminate repeated explanations that appear in multiple places
   - Consolidate overlapping concepts into single, clearer statements

2. **Avoid Too Specific Cases**:
   - Replace overly specific examples with generalizable patterns
   - Convert hardcoded values to placeholders (e.g., [TARGET], [QUERY])
   - Delete task-specific details or specific cases that don't apply to similar problems

3. **Logical Consolidation**:
   - Merge workflows that share substantial overlap into variants
   - Extract common preliminary steps into dedicated sections
   - Group related tool templates and query patterns together
   - Consolidate similar pitfalls into broader categories

4. **Format Optimization**:
   - Ensure consistent structure and formatting throughout
   - Improve section hierarchy and logical flow
   - Make workflows easier to scan (clear steps, consistent formatting)
   - Organize content from general principles → specific techniques

5. **Content Quality**:
   - Keep description concise and focused on core purpose
   - Ensure all content is actionable and reusable
   - Remove verbose explanations that don't add value
   - Maintain the most essential and distinctive elements

### Principles:
- Prioritize generalizability over specificity
- Keep what enables reuse across similar problems
- Remove what only applies to one particular case
- Maintain clarity and actionability

Output ONLY the refined SKILL.md starting with `---`. No preamble.

```
<current_skill>
  {skill_content}
</current_skill>
```

## C.2.3. SKILL ADAPTATION PROMPTS

---

**ADAPT_SKILL_PROMPT**

You are an expert agent assistant. Tailor the general skill to this specific task.

**Your Goals:**

1. **Select What's Relevant**: Look at the task and images - which parts of the skill actually apply here? Remove everything else.

2. **Integrate Experiences**: If experiences are provided, weave their insights into the relevant workflow steps. They often contain practical tips that complement the skill's structure.

3. **Keep Templates Ready**: Preserve any code templates or query patterns that might be useful, but you can adjust placeholders to be more task-relevant.

4. **Stay Lean**: The adapted skill should be a focused guide, not a comprehensive manual. ∼400 words max.

**Input:**
```
<base_skill>{base_skill}</base_skill>
<experiences>{experiences}</experiences>
<task>{task}</task>
```
**CRITICAL:** Output a reusable methodology guide, NOT a pre-filled answer. Use placeholders (e.g., "the observed value", "extracted number") instead of actual data from images or task.

Output ONLY the adapted skill content (markdown format starting with #). Do NOT include frontmatter metadata (no ---
blocks with name/description/version). Focus on what will actually help solve this task.

---

**SKILL_INJECTION_HEADER**

Here are practical experiences and skills for tool-based visual reasoning:
```
<skill>
{skill_content}
</skill>
```
You can use it as reference if it is relevant to help you solve the problem. You can also have your own ideas or other approaches.
Your instruction is following:

---

## C.2.4. EXPERIENCE GENERATION PROMPTS

---

**ROLLOUT_SUMMARY**

You are a World-Class reasoning analysis expert. A multimodal agent system uses tool-based visual reasoning to solve the given problem. The agent may have been provided with some experiences. Please summarize the following trajectory (also called rollout) step-by-step:

**Summarization Guidelines:**

1. **For each turn or step:**

   - Describe which tool was used and with what parameters, explain the reasoning for this specific action, and note which experience (if any) was applied and how.
   - If this turn was part of meta-reasoning skills: identify the meta-reasoning type (e.g., question decomposition, sequential reflection, self-correction, self-verification, etc.) and explain how its outcome influenced subsequent steps or the final result.

2. Given the trajectory and the correct answer, identify and explain any steps that represent detours, errors, backtracking, or any other failure patterns, highlighting why they might have occurred and what their impact was on the trajectory's progress. Discuss how the agent's tool-using knowledge and meta-reasoning skills handled or mitigated these issues.

3. Maintain all the core outcomes of each turn or step, even if it was part of a flawed process.

4. **Thinking with images actions** (if intermediate images are provided):

   - Document any image preprocessing operations (cropping, filtering, enhancement, etc.) or image searching (searching the web for relevant images, etc.) operations, these operations generated intermediate images. You need to analyze and note their impact.
   - For intermediate images that were generated and used: identify which visual features were extracted and how they can help the agent to reason better.
   - Suggest specific visual operations that could improve reasoning. If no intermediate images were generated and used in some rollout steps, but could have been helpful, note this as a potential point of improvement.

**Input:** `<trajectory>{trajectory}</trajectory>`
Provide a clear, structured summary of the trajectory.

---

## CROSS_ROLLOUT_CRITIQUE

You are a reasoning analysis expert. Review these problem-solving attempts and extract practical lessons that could help future attempts.
**Analysis Framework:**
*1. Trajectory Review:*

- What worked? What key decisions or insights led to correct answers?

- What didn't work? Where did reasoning go wrong, and why?

- Were any provided experiences missed or not helpful enough?

- How were different tools combined? What sequences were effective?

- For visual operations: What preprocessing helped? What was missing?

*2. Experience Extraction:*
Create experiences in two categories:

- **Execution Tips**: Practical advice on tool usage - when to use which tool, what parameters work best, how to interpret results effectively.

- **Decision Rules**: Simple guidelines for reasoning choices - when to decompose a problem, when to search for information, when to double-check results.

You have two options: [add, modify]

- **add**: A genuinely new lesson not covered by existing experiences.

- **modify**: Improve an existing experience (reference its ID) if you find it could be more accurate, clearer, or more actionable based on this trajectory.

The `<experiences_used>` section below shows the experiences that were used to guide this sample. Review them against the trajectory outcomes to identify potential improvements.
You can apply at most {max_ops} updates for this case.
*3. Experience Quality:*

- Keep each experience under 64 words.

- Start with the situation or condition when the advice applies.

- Make it general enough to apply to similar problems.

- Focus on actionable guidance, not abstract principles.

- Avoid specific examples - the experience should generalize.

Provide detailed reasoning following the above framework, then conclude with:

```
[
  {"option": "add", "experience": "the new generalizable experience"},
  {"option": "modify", "experience": "the modified experience",
   "modified_from": "E17"},
  ...
]
```

Note: You may use only one type of update. Quality over quantity.
**Input:**
`<question>{question}</question>`
`<summaries>{summaries}</summaries>`
`<experiences_used>{experiences}</experiences_used>`
`<ground_truth>{groundtruth}</ground_truth>`

---

## MERGE_PROMPT

You are an experience library management expert. Merge the following experiences into a single, comprehensive experience.
**Experiences to merge:**
{experiences_text}

**Requirements:**

1. Contain all important information points from all experiences

2. Be clear, generalizable, and no more than 64 words

3. Maintain core lessons and decision points

4. Avoid redundancy while preserving unique insights

Output ONLY the merged experience text, no other text or comments.

---

## EXPERIENCE_MANAGE_PROMPT

You are an experience library curator. The library has grown through multiple batches and may contain redundancy. Perform a global refinement pass.

**Current Library Size:** {exp_count} experiences

**Target:** Reduce to 80-100 high-quality, diverse experiences

**Refinement Goals:**

1. **Merge Truly Redundant**: Combine experiences expressing the same core insight.

2. **Generalize Over-Specific**: Abstract experiences tied to specific scenarios into general patterns.

3. **Delete Low-Value or Too-Specific**: Remove experiences that are too obvious or rarely applicable, or too specific and only applicable to a single case or scenario.

**Operations:**

- **merge**: Combine 2+ experiences into one (provide merged_from list)

- **delete**: Remove an experience entirely (provide deleted_id)

**Quality Standard:**

- Under 64 words, clear and actionable, and generalizable to a class of problems.

- Starts with the trigger condition ("When...", "For...", "If...")

- No specific examples or object names unless essential

**Important:**

- Only act on clear redundancy or low quality. Preserve diversity.

- If the library is already well-curated, minimal changes are fine.

Output your reasoning, then:

```
[
  {"option": "merge", "experience": "text",
   "merged_from": ["E12", "E23"]},
  {"option": "delete", "deleted_id": "E45"},
  ...
]
```

**Input:** <experiences>{experiences}</experiences>

---

## INJECTION_HEADER

Here are practical tips for tool-based visual reasoning, gathered from similar problems:
{bullets}
These experiences highlight common patterns and pitfalls. When you encounter matching situations, consider applying the suggested approaches. You can reference them by ID (e.g., [E12]) in your reasoning.
Your instruction is following:

## C.2.5. EXPERIENCE ADAPTATION PROMPTS

---

**TASK_DECOMPOSITION_PROMPT**

You are an Expert Visual Reasoning Strategist. Your objective is to deconstruct a complex visual task into 2-3 distinct, actionable subtasks to retrieve the most relevant methodological guidance from the experience library.
**Task:** {task_description}
The experience library contains abstract methodological guidelines for visual reasoning tasks.
**For each subtask, generate a JSON object with:**

- **Type**: A concise category describing the methodological phase.
- **Query**: A methodology-focused search query designed to retrieve "how-to" guides or best practices, based on current task description and image details.

**The query should target one of the following aspects:**

1. Tool Utilization: Best practices for using specific tools (e.g., Code Interpreter, Web Search, Image Search) effectively.

2. Reasoning Strategy: Reasoning frameworks or potential solution paths that can be used to solve the task.

3. Challenge Mitigation: Techniques to handle anticipated challenges (e.g., "handling flipped images," "small objects in the image").

**CRITICAL:** The query must abstract away from specific image details (like "red apples") and focus on the underlying *technical challenge* (like "color-based object filtering").
**Required output format:**

```
[
  {"type": "visual_extraction",
   "query": "Techniques for analyzing..."},
  {"type": "logic_synthesis",
   "query": "Frameworks for mapping..."}
]
```

Output ONLY the complete and valid JSON array, no additional text.

---

**EXPERIENCE_REWRITE_PROMPT**

You are an expert AI mentor adapting retrieved methodological experiences to strictly fit a specific visual reasoning task.
**Task:** {task_description}
**Retrieved experiences:** {experiences_text}
Rewrite these experiences to provide cohesive, unified operational guidance applicable to the task and its images. For each experience, strictly adhere to the following rewriting guidelines:

1. **Operational Focus**: Transform abstract descriptions into specific, actionable execution tips (e.g., tool-using actions, error handling, etc.). Focus on detailed operations and practical techniques rather than high-level summaries.

2. **Pitfalls & Best Practices**: Explicitly integrate common pitfalls to avoid and best practices to follow based on the experience.

3. **Contextual Adaptation**: Keep the core methodological insights but adapt the language to be directly relevant to the current visual task context. Do not make it *too* specific (overfitting), but ensure it is practically useful.

4. **Tone**: Use clear, constructive, and suggestive language (e.g., "Consider checking...", "It is effective to...") rather than direct commands.

If an experience is irrelevant or redundant, you may delete it by not outputting it in the JSON object. Only focus on the experiences that can contribute to solving the task.
Output ONLY a valid JSON object mapping experience IDs to the rewritten guidance text. Strictly follow the format:

```
{
  "id1": "rewritten experience 1",
  "id2": "rewritten experience 2",
  ...
}
```

---

# D. Case Studies

We present qualitative case studies demonstrating how XSKILL guides agents to adopt more systematic and objective reasoning strategies. We examine two scenarios: (1) XSKILL vs. tool-only baseline, and (2) ablation analysis comparing

Skill+Experience, Skill-only, and Experience-only configurations.

### D.1. Knowledge Base Examples

To illustrate the concrete structure of our dual-stream knowledge representation, we provide examples of accumulated Skills and Experiences used during inference. These knowledge artifacts are generated by $MLLM_{kb}$ during Phase I (Accumulation) and retrieved/adapted during Phase II (Inference).

#### D.1.1. SKILL EXAMPLE: VISUALLOGICARCHITECT

The following skill generated by Gemini3-Flash on VisualToolBench demonstrates the task-level guidance provided to agents. It is stored as a Markdown document with metadata (name, description, version), structured workflows, and reusable tool templates.

---

**Skill Document: VisualLogicArchitect.md (v20.0.0)**

**Metadata:**
- **Name:** VisualLogicArchitect
- **Description:** A unified framework for multi-domain visual analysis: quantitative document auditing (financial/TAT), network & geometric pathfinding, scientific-clinical diagnostics (NMR/Survival), and tactical spatio-temporal scenario analysis (sports/market charts).
- **Version:** 20.0.0

**When to Use:**
- **Quantitative & Document Auditing**: Verifying financial grids (Sankey, income statements), receipts, or clinical reports for Turnaround Time (TAT) and accreditation compliance.
- **Network & Geometric Systems**: Solving circuits, transit routing, and analyzing radial gauges or lever physics.
- **Scientific & Clinical Diagnostics**: Analyzing NMR spectra, genomic plots, and Kaplan-Meier (KM) survival curves for median outcomes and milestone benefits.
- **Tactical & Spatio-Temporal Scenarios**: Determining game outcomes (sports), market positions, or performing "what-if" financial dashboard re-sequencing.

**Strategy Overview:**
1. **Normalization & Orientation**: Detect reversed text or axes. Use Python to flip/rotate until legible. Decouple physical layout from mathematical grid.
2. **Spatio-Temporal Re-sequencing**: For motion or flow, reconstruct the "just before" and "just after." Use shadow positions to anchor true locations.
3. **Segmentation & ROI Census**: Isolate Regions of Interest (ROI). Use systematic scans for dense markers to prevent double-counting.
4. **Logical Reconciliation**: Apply domain rules (Ohm's Law, NMR DEPT logic, Market Span Rule, Clinical Accreditation Standards).

**Workflow 1: Financial, Flow & Document Auditing**
1. **Data Census & Extraction**: Identify the "Grand Total" node or "Receipt Total." For Sankey/Financial diagrams, identify sub-contributors and "Exclusion" nodes. For TAT, record timestamps.
2. **Relevant Base Calculation**: Calculate the denominator by subtracting excluded values or summing specific sub-categories.
3. **Metric Ranking & Synthesis**: Sort contributors by growth/value to find top N performers. For TAT Analysis, use `web_search` for accreditation benchmarks. Recalculate totals from raw line items.

**Workflow 2: Geometric, Radial & Network Systems**
1. **Coordinate & Landmark Mapping**: Identify origin and axis directions. In inverted systems, "mathematically highest" may be physically at the bottom.
2. **Path Efficiency Audit**: Map landmarks to nearest station. Compare Hub (1 transfer) vs. Crosstown (2+ transfers) routes. If a 2-transfer route has ≥25% fewer stops, it is likely fastest.
3. **Geometric Extraction**: For radial paths, anchor on center and move outward to prevent line jumping. Calculate lever arms relative to fulcrum.

**Tool Templates:**
**Python: Normalization & Quantitative Logic**

---

```
from PIL import Image, ImageOps, ImageEnhance
from datetime import datetime

def process_visual(img_path, rotate=0, flip_h=False, crop_norm=None):
    img = Image.open(img_path)
    if flip_h: img = ImageOps.mirror(img)
    if rotate: img = img.rotate(rotate, expand=True)
    if crop_norm: # [ymin, xmin, ymax, xmax] 0-1000
        w, h = img.size
        img = img.crop((crop_norm[1]*w/1000, crop_norm[0]*h/1000,
                        crop_norm[3]*w/1000, crop_norm[2]*h/1000))
    return img

def calculate_duration(start_str, end_str, fmt="%d/%m/%Y %H:%M"):
    start = datetime.strptime(start_str, fmt)
    end = datetime.strptime(end_str, fmt)
    delta = end - start
    return delta.total_seconds() / 3600 # Returns hours
```

**Watch Out For:**

- **The Perspective Trap**: A ball in mid-air often appears "past" a line. Always trust the shadow's contact point.

- **The Weekend Trap**: TAT standards often use "Business Days." Check if the accreditor counts Saturdays.

- **Unit Divergence**: Dashboards may mix Millions (MUSD) and absolute USD. KM plots mix months (X-axis) and percentages (Y-axis).

- **Mirrored Digits**: In reversed images, '2' vs '5' or '6' vs '9' are easily confused. Flip before reading.

### D.1.2. EXPERIENCE EXAMPLES

The following experiences generated by Gemini3-Flash on VisualToolBench demonstrate action-level tips extracted from successful rollouts. Each experience is stored as a JSON entry with a condition-action pair and semantic embedding (not shown). We present 15 representative examples from the accumulated knowledge base.

---

**Experience Entries: experiences.json (Sample)**

**E0:** In complex structured problems, identify 'zero-impact' zones where data does not influence the final outcome. Recognizing these regions (like the 'zero-zone' in block-triangular matrices) allows you to ignore irrelevant variables or noise, streamlining the reasoning process and minimizing the potential for errors in data extraction and calculation.

**E1:** In sports analytics, explicitly distinguish between the visual context (e.g., a Finals game) and the requested statistical period (e.g., Regular Season). Include the specific period in search queries to ensure retrieved data matches the question's requirements, as player performance often differs significantly between tournament stages.

**E2:** To confirm a character is truly absent, apply brightness and contrast enhancements specifically to 'shadow traps'—dark or cluttered background areas. Only declare a character missing after a high-magnification scan of these enhanced regions, as small silhouettes or background cameos are easily missed in standard views.

**E5:** Before interpreting trends or identifying the "current" price, verify the timeline using axis labels (e.g., dates). If dates decrease from left to right, the image is mirrored and requires a horizontal flip. This ensures the "latest" data point is correctly identified and indicators are interpreted in the correct temporal sequence.

**E7:** When selecting items based on one metric (e.g., growth rate) to perform calculations with another (e.g., revenue), extract both values for all candidates into a structured list. This prevents 'metric confusion' errors where the ranking value is accidentally used in the final arithmetic instead of the required target value.

**E22:** Normalize rotated, mirrored, or distorted layouts using code (ImageOps.mirror, homography) to restore text flow and prevent character hallucinations. Use reference points to verify legibility. Post-transformation, re-verify coordinate systems and origin shifts using wide-view crops. Define spatial terms mathematically and use high-resolution ROI crops to ensure precise extraction of small text and numerical labels for accurate calculations.

**E26:** When filtering items by category in a list, perform a 'Categorical Exhaustion' scan: after identifying obvious matches, review every remaining item to ensure no relevant entries were missed due to abbreviations, synonyms, or generic naming. This is vital for accurate summation in receipts where item names are often truncated or non-obvious.

**E33:** For images with perspective distortion or structural analysis, establish a coordinate system using on-image anchors. Before performing kinematic, spatial, or force-balance calculations, check for geometric and load symmetry to simplify reactions proportionally. Calibrate pixel-to-unit ratios and verify mathematical signs relative to reference axes to ensure geometric accuracy across all calculations.

**E46:** When performing classification based on visual markers, implement a 'mandatory feature check.' Before finalizing a category, verify the presence of all required secondary characteristics and the absence of contradictory ones. This cross-verification prevents a single visual hallucination or misidentification from causing a cascading reasoning failure for the entire problem.

**E53:** When tool-generated descriptions (captions, OCR, or detections) conflict with visual evidence, prioritize direct inspection of the raw image. Use high-magnification crops to verify stances, object possession, or text segments. This prevents 'hallucination propagation,' where a tool's initial error misleads the reasoning chain and results in an incorrect final answer.

**E61:** To locate facilities or segments on dense maps, establish "spatial anchors" using secondary data like addresses, intersections, elevation profiles, or legends. Identifying precise icons or start/end points first enables you to ignore irrelevant sections and focus visual search or high-resolution cropping on coordinates strictly meeting task criteria, preventing inclusion errors near boundaries.

**E70:** For complex spatial reasoning, use Python to draw explicit geometric overlays (lines, rays) on the processed image. Visualizing a path (like a Bishop's diagonal) as an explicit layer helps identify intersections with small or low-contrast objects that are difficult to track mentally across a cluttered or dark background.

**E78:** When dealing with low-contrast, dark, or distorted images, use Python (e.g., CLAHE, brightness/contrast enhancement, mirroring) to isolate and clarify faint features like small text or decimal points. Compare enhanced crops with the original to ensure features are real. This prevents OCR errors and misidentification of critical numerical or status indicators.

**E82:** In sports analytics, use scoreboards, parity rules (e.g., even/odd service), and current scores as logical anchors to determine game states and player roles. Cross-reference visual evidence with domain-specific rules to predict transitions or identify active players, especially when motion blur or camera angles make direct visual identification difficult.

**E88:** For complex quantitative tasks, perform a 'Balance Check' by verifying printed totals against summed components. If source data contains error flags or is blurry, aggregate raw line items manually and use chronological order to resolve ambiguities. Reconcile algebraic results against visual evidence to identify early-stage reasoning errors.

## D.2. XSKILL vs. Tool-Only Baseline

### D.2.1. TASK DESCRIPTION

**Question:** "In what color area is the second 'GOOD' text on the car body located?"

**Ground Truth:** Purple

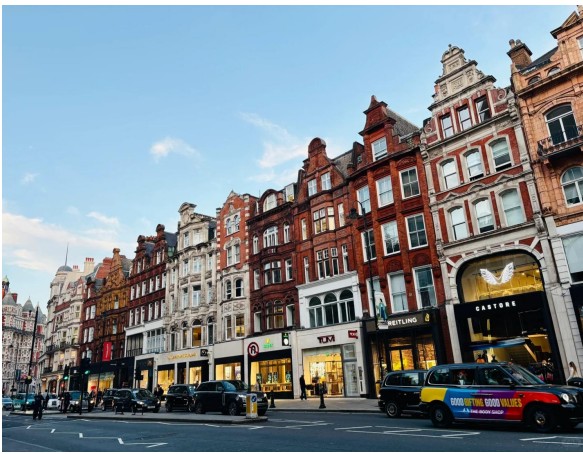

*Figure 6.* Input image showing a street scene with a taxi displaying "GOOD GIFTING GOOD VALUES" text on its side.

This task requires (1) locating and counting text instances on a small, partially visible object, and (2) identifying the background color objectively despite lighting variations and visual ambiguity.

D.2.2. CASE 1: DIRECT ANSWER VS. SYSTEMATIC ANALYSIS (ROLLOUT 2)

| With XSKILL (Correct: Purple) | Without XSKILL (Incorrect: Red) |
|---|---|
| *Phase 1: Injected Knowledge* | |

**Injected Skill Fragment:**
- *Systematic Text Instance Location*: Establish positional anchor, sequential scan & enumeration, neighbor verification
- *Objective Color Identification*: Isolate ROI, forensic pixel sampling, color determination

**Tool Template Provided:**
```
def forensic_crop(img, box):
    # box: [ymin,xmin,ymax,xmax] (0-1000)
    w,h = img.size
    crop = img.crop(...)
```

*No experiences or skills injected.*
*Agent relies solely on visual inspection and base reasoning capabilities.*

*Phase 2: Agent Reasoning & Execution*

**Turn 1 - Reasoning:**
1. Locate text: Find "GOOD GIFTING GOOD VALUES"
2. Identify second "GOOD": Scan left-to-right
3. **Apply skill**: Isolate ROI with bounding box
4. **Objective verification**: Use code to sample pixels

**Tool Call:** `code_interpreter`
```
box = [835, 840, 865, 900]
cropped_roi = original_image.crop(
    (left, top, right, bottom))
avg_color = pixels.mean(axis=0)
# RGB: (34, 41, 52) → Blue
```

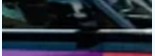

**Turn 2 - Final Answer:**
**"The second 'GOOD' text is located in the purple area."**

**Turn 1 - Direct Answer:**
*"The second 'GOOD' text on the car body is located in the **red** color area."*

*No tool calls made.*
*No objective verification performed.*

**Incorrect** - Relied on subjective visual judgment without systematic analysis.

**Key Insights**

**(1) Objective Verification:** The injected skill guides the agent to objectively verify colors through pixel analysis rather than relying on subjective visual inspection. The tool-only agent immediately provides an incorrect answer ("red") based on visual appearance, while XSKILL prompts systematic ROI extraction and forensic color analysis, leading to the correct answer ("purple").

D.2.3. CASE 2: ITERATIVE REFINEMENT WITH COLOR SPACE CONVERSION (ROLLOUT 3)

This case demonstrates XSKILL's ability to guide more sophisticated analytical strategies, including iterative refinement and advanced color space analysis.

| With XSKILL (Correct: Purple) | Without XSKILL (Incorrect: Red) |
|---|---|
| Phase 1: Knowledge Injection (Same as Case 1) | |
| *Same skill and tool template injected as Case 1, demonstrating consistent guidance across different rollouts.* | *No knowledge injected.* |
| Phase 2: Agent Reasoning & Execution | |

**Turn 1 - Initial Analysis:**
1. Apply systematic text location workflow
2. Define precise bounding box [865,800,895,870]
3. Forensic pixel sampling via code

**Tool Call:** `code_interpreter`

```
box = [865, 800, 895, 870]
avg_color = np.mean(roi_array, axis=(0,1))
# RGB: (84, 59, 72)
# Result: "Undetermined"
```

**Turn 2 - Iterative Refinement:**
*Recognizing ambiguity, agent refines approach:*
1. Adjust bounding box [860,800,880,870]
2. **Convert RGB to HSV** for robust color ID

**Tool Call:** `code_interpreter`

```
h, s, v = colorsys.rgb_to_hsv(r,g,b)
h_deg = h * 360  # Hue in degrees
# RGB: (54, 38, 91)
# HSV: Hue=258.1°, S=0.58, V=0.36
# → Purple/Violet (240-300°)
```

**Turn 3 - Final Answer:**
**"The second 'GOOD' text is located in a purple area."**
*Confirmed via HSV analysis: Hue=258.1° (purple/violet range).*

**Turn 1 - Direct Answer:**
*"The second 'GOOD' text on the car body is located in the red color area."*

*Identical behavior to Case 1: No tools used, no verification performed.*

**Incorrect** - Same subjective judgment error.

---

**Key Insights**

**(1) Consistency:** XSKILL demonstrates consistent improvement across different rollouts, while the tool-only baseline repeatedly makes the same error.
**(2) Sophisticated Strategies:** The injected skill enables more advanced techniques beyond basic workflows. When initial RGB analysis yields ambiguous results, the agent autonomously refines its approach by converting to HSV color space for robust color identification.
**(3) Iterative Problem-Solving:** XSKILL encourages systematic iteration: when uncertain, adjust parameters and apply complementary analysis methods rather than settling for ambiguous results.

