# OpenReview forum: "XSkill: Continual Learning from Experience and Skills in Multimodal Agents"
_ICML.cc/2026/Conference — ICML 2026 regular_

### Official Review · Reviewer_oTwA · 2026-03-11

**Soundness:** 2
**Presentation:** 3
**Significance:** 2
**Originality:** 2
**Overall Recommendation:** 4
**Confidence:** 3

**Summary:**

This paper introduces EXSKILL, a framework designed to enable continual learning in multimodal agents through external knowledge accumulation. The core idea is to build a knowledge base consisting of two complementary components: skills, which encode structured workflows and tool usage patterns at the task level, and experiences, which capture more fine-grained action-level guidance extracted from agent trajectories.

During training, the system executes multiple rollouts for each task and summarizes the resulting trajectories to extract skill fragments and experience entries. These pieces of knowledge are then consolidated and stored in a shared knowledge base. At inference time, EXSKILL decomposes a new task into sub-tasks, retrieves relevant experiences, rewrites them according to the current visual context, and adapts global skills into task-specific guidance that is injected into the agent prompt.

Experiments on several multimodal agent benchmarks show improvements over tool-only agents and a few memory-based baselines. The paper also includes ablations analyzing the role of skills, experiences, and rollout-based accumulation.

**Compliance With Llm Reviewing Policy:**

Affirmed.

**Key Questions For Authors:**

1. How sensitive is the performance to the number of rollouts used during the accumulation phase?
2. Can the authors provide a more direct analysis of the extracted skills and experiences (e.g., redundancy, accuracy, or coverage)?
3. How does the computational cost of EXSKILL compare with simpler agent pipelines?
4. What factors determine whether accumulated knowledge can transfer across different backbone models?

**Limitations:**

The proposed framework relies heavily on a multi-stage pipeline involving several large language model components, which may limit its practicality in real-world deployments. In addition, the effectiveness of the approach depends on the quality of trajectory summarization and knowledge extraction, which may introduce noise into the knowledge base. Finally, the current experiments suggest that the accumulated knowledge does not always generalize well across different models or domains.

**Strengths And Weaknesses:**

Strengths:

1. The paper tackles an important problem. Current multimodal agents typically treat each task independently, and the idea of accumulating reusable knowledge across tasks is well motivated.

2. The distinction between skills and experiences is intuitive. Separating task-level workflows from action-level heuristics provides a reasonable abstraction for organizing agent knowledge.

3. The framework considers visual context during knowledge extraction. Unlike many previous memory-based approaches that focus mainly on textual reasoning traces, this work explicitly incorporates visual observations when summarizing trajectories.

4. The experimental section is fairly thorough. The paper evaluates the method on several benchmarks and includes ablation studies examining different components of the system.

Weaknesses:

1. The overall novelty appears somewhat limited. The framework mainly combines existing ideas such as trajectory summarization, external memory, and retrieval-augmented prompting. While the skill/experience distinction is reasonable, the conceptual contribution beyond existing agent knowledge-base systems is not entirely clear.

2. The system pipeline is quite complex. EXSKILL relies on multiple LLM modules and several stages of processing (summarization, critique, consolidation, retrieval, rewriting, adaptation). This introduces significant computational overhead, yet the paper does not discuss efficiency or scalability in much detail.

3. The quality of extracted knowledge is not directly evaluated. The method assumes that trajectory summaries can reliably produce useful skills and experiences, but there is little analysis of the correctness or redundancy of the resulting knowledge base.

4. The evaluation could include stronger baselines. Most comparisons are with earlier memory-based agent approaches. It would be helpful to compare with more recent multimodal agent frameworks or planning-based agents.

---

> ### Author Rebuttal · Authors · 2026-03-31
>
> We thank the reviewer for the thoughtful feedback and for recognizing the reasonableness of our skill and experience distinction. We address your questions below.
>
> ### R1: Sensitivity of Performance to Rollout Quantity
>
> > Question-1: How sensitive is the performance to the number of rollouts used during the accumulation phase?
>
> We have analyzed rollout quantity in Section 3.4 and Figure 3. As N increases, Average@4 and Pass@4 show consistent upward trends.
>
> ### R2: Complexity and Novelty of the System Pipeline
>
> > Weakness-1 & Weakness-2: The overall novelty appears somewhat limited... The system pipeline is quite complex.
>
> The dual-stream memory design, separating task-level Skills from action-level Experiences, is our primary conceptual contribution. This architecture maps to procedural and episodic memory, respectively, addressing the integration gap in prior agent frameworks. By automatically generating skills, EXSKILL introduces a new paradigm that moves beyond manual expert authoring.
>
> Regarding complexity, EXSKILL is a cohesive system where visual grounding and adaptation function in synergy with the dual-stream memory. Grounding anchors insights in visual states, and adaptation applies them to new contexts. This coordination is essential for multimodal reasoning, where simple trajectory recording often fails. The multi-stage pipeline is not a collection of bolted-on parts but a unified design that enables the significant performance gains reported.
>
> ### R3: Quality Verification of Extracted Knowledge
>
> > Weakness-3 & Question-2: The quality of extracted knowledge is not directly evaluated... Can the authors provide a more direct analysis?
>
> To evaluate knowledge quality, we conduct a double-blind experiment where a strong LLM judge (Gemini-3-Pro) evaluates the memory from EXSKILL and baselines, scoring on a 1–5 scale. The assessment follows a uniform rubric including accuracy, trigger specificity, actionability, and generalization.
>
> |Method|Accuracy|TrigSpec|Action|Gen|
> |---|---|---|---|---|
> |AWM|3.64|2.96|3.22|3.34|
> |DC|3.55|3.74|3.42|3.46|
> |Agent-KB|3.76|3.72|3.54|3.58|
> |EXSKILL|3.88|4.12|4.06|3.74|
>
> Knowledge extracted by EXSKILL scores significantly higher in actionability and trigger specificity than baselines. Baselines often extract vague text heuristics, while EXSKILL clears redundant information through hierarchical consolidation.
>
> ### R4: Computational Cost and Efficiency
>
> > Question-3: How does the computational cost of EXSKILL compare with simpler agent pipelines?
>
> While Phase I knowledge accumulation incurs higher initial overhead than simpler methods, this investment is both reusable and amortizable. Due to the dual-stream architecture, model calls per rollout are roughly twice those of DC or AWM and 1.4 times those of Agent-KB.
>
> However, such structured organization is essential in multimodal reasoning. Simple trajectory recording like AWM and DC often yields low-quality knowledge that can actually degrade performance (See Table 2). During Phase II inference, EXSKILL maintains efficiency with model calls consistent with Agent-KB. This upfront accumulation pays off through superior zero-shot transfer and execution accuracy, delivering 10% to 30% gains across benchmarks. We consider this accuracy-cost trade-off highly favorable for complex tasks. Furthermore, EXSKILL allows for flexible budget control by adjusting retrieval top-k or update frequency. We include a detailed discussion of these computational trade-offs in the final manuscript.
>
> ### R5: Cross-Backbone Model Transfer
>
> > Question-4: What factors determine whether accumulated knowledge can transfer across different backbone models?
>
> Cross-model knowledge transfer depends on two factors. First, source trajectory quality is critical. Stronger models like Gemini-3-Flash generate successful trajectories whose knowledge benefits weaker models like GPT-5-mini. Second, the receiving model needs sufficient baseline capability. As shown with Qwen3-VL in Appendix A, if the receiving model is too weak, complex knowledge may interfere with native tool-use and lower performance. Success requires a basic reasoning foundation to execute provided strategies.
>
> ### R6: Comparison with Recent Multimodal Agent Baselines
>
> > Weakness-4: The evaluation could include stronger baselines... compare with more recent multimodal agent frameworks.
>
> We add comparisons with recent multimodal agent frameworks on MMSearch-Plus.
>
> |Setting|Performance|
> |---|---|
> |Gemini-2.5-Pro (Base)|21.56%|
> |RedSearcher-MM (SFT + RL)|26.60%|
> |Vision-DeepResearch (SFT + RL)|28.50%|
> |EXSKILL+Gemini-2.5-Pro|27.96%|
>
> Without expensive RL training, EXSKILL enables Gemini-2.5-Pro to achieve performance comparable to or exceeding top-tier search agents trained with SFT and RL. This demonstrates the effectiveness of our non-parametric framework.
>
> We thank the reviewer for prompting these critical clarifications and experiments, which undoubtedly make the work more solid.

---

### Official Review · Reviewer_gRAd · 2026-03-12

**Soundness:** 3
**Presentation:** 3
**Significance:** 3
**Originality:** 2
**Overall Recommendation:** 4
**Confidence:** 3

**Summary:**

The paper proposes EXSKILL, a framework for enabling continual learning in multimodal agents through experience accumulation without updating model parameters. The key idea is to store two complementary types of knowledge extracted from agent trajectories: task-level Skills (structured workflows and tool templates) and action-level Experiences (context-specific tactical insights). The framework automatically extracts these using visually grounded summarization that integrates visual observations with textual reasoning. It further introduces hierarchical consolidation to maintain knowledge quality and diversity, and context-aware adaptation to tailor retrieved knowledge to the current visual context. Experiments on five multimodal benchmarks involving visual tool use and multimodal search show consistent improvements over existing experience-based agent memory systems.

**Compliance With Llm Reviewing Policy:**

Affirmed.

**Final Justification:**

My concerns are basically addressed, and I will maintain my initial justification.

**Key Questions For Authors:**

-How sensitive is the performance to the size of the accumulated experience/skill repository? Does performance plateau or degrade as the memory grows larger?

-Can the authors provide an ablation isolating the impact of skills vs. experiences to better quantify the contribution of the dual-memory design?

-How does visually grounded summarization differ from purely text-based trajectory summarization in practice, and what is its measurable impact on downstream performance?

-What are the computational and storage costs of maintaining and retrieving from the hierarchical knowledge base during long-term deployment?

**Limitations:**

yes

**Strengths And Weaknesses:**

Strengths

+The paper tackles an important limitation of current multimodal agents—lack of effective experience reuse across tasks—and proposes a clear framework for continual improvement without parameter updates.

+The distinction between task-level skills and action-level experiences is intuitive and provides a practical way to structure agent memory.

+The proposed visually grounded summarization is a reasonable step toward incorporating visual observations when extracting reusable knowledge, which many prior approaches ignore.

+Empirical evaluation spans multiple benchmarks and shows consistent improvements over several recent agent memory baselines.

Weaknesses

-The novelty is somewhat incremental: the approach mainly combines existing ideas from agent memory, trajectory summarization, and retrieval-based adaptation into a unified framework.

-The experimental gains are relatively modest, and it is unclear how much improvement comes from the dual memory design versus other components.

-The scalability of the experience/skill repository over long-term deployment is not fully discussed, particularly regarding storage growth and retrieval efficiency.

-Some methodological details (e.g., how visual grounding specifically influences the extracted knowledge or affects retrieval) could be more clearly analyzed.

---

> ### Author Rebuttal · Authors · 2026-03-31
>
> We thank you for recognizing that our dual-stream framework structures agent memory in an intuitive, practical way. We address your questions below.
>
> ### R1: Innovative Contribution of the Overall Framework
>
> > Weakness-1: The novelty is somewhat incremental: the approach mainly combines existing ideas... into a unified framework.
>
> The dual-stream memory design is our primary conceptual contribution and represents a methodological innovation. Cognitively, Skills function as procedural memory, while Experiences function as episodic memory. Most prior work focuses on one memory type for agents and neglects integrating both. Furthermore, as Skills become indispensable in agent harnesses and are mostly manually authored, EXSKILL explores automatic generation, storage, and utilization of skills.
>
> This unified framework is not a simple collection of techniques. Visual grounding anchors extracted insights in actual visual states, while adaptation applies them to unseen contexts. These components work in synergy to overcome the limitations of text-only trajectory logs, making the framework uniquely suited for multimodal agents. This synergy is what enables the performance gains reported in our experiments.
>
> ### R2: Sensitivity and Scalability of Accumulated Knowledge Base
>
> > Weakness-3 & Question-1: The scalability of the experience/skill repository over long-term deployment is not fully discussed.
>
> To address sensitivity to knowledge base size, we track performance during continuous accumulation. As the agent processes more tasks and enriches its memory, overall performance (Average@4) shows a consistent upward trend rather than a decline. This indicates that our hierarchical integration effectively handles growing knowledge:
>
> |Accumulation Tasks|0|20|40|60|80|100|
> |---|---|---|---|---|---|---|
> |Gemini-2.5-Pro (%)|25.35|25.12|26.70|28.30|31.30|30.49|
> |Gemini-3-Flash (%)|41.94|42.76|44.86|44.28|45.79|46.50|
>
> Crucially, the knowledge base does not grow indefinitely. As described in Section 2.2.3, the Experience and Skill Managers utilize a hierarchical integration mechanism. They actively merge semantically redundant entries and delete low-quality or overly specific knowledge. This dynamic filtering can reduce performance degradation significantly during long-term deployment.
>
> ### R3: Ablation Study on Skills and Experiences
>
> > Weakness-2 & Question-2: The experimental gains are relatively modest, and it is unclear how much improvement comes from the dual memory design.
>
> We have reported this ablation in Section 3.3 of the submitted manuscript. Our main findings are (1) Removing Experiences or Skills both lowers the performance. (2) Skills significantly reduce execution errors, while Experiences shape strategic decisions and tool selection.
>
> ### R4: Impact of Visual Grounding on Knowledge Extraction
>
> > Question-3 & Weakness-4: How does visually grounded summarization differ from purely text-based trajectory summarization in practice?
>
> To measure the impact of visual grounding, we introduce a new ablation setting, EXSKILL (Text-Only), where Phase I knowledge extraction relies strictly on text trajectory logs.
>
> Quantitative results on VisualToolBench (Gemini-2.5-Pro):
>
> |Setting|Avg@4|Delta|
> |---|---|---|
> |w/ Tools|25.35%|-|
> |EXSKILL(Text-Only)|27.57%|+2.22%|
> |EXSKILL(Full)|30.49%|+5.14%|
>
> Text-only extraction provides limited improvement, while the full visual grounding pipeline delivers a substantial gain. We also use a strong LLM judge (Gemini-3-Pro) for double-blind scoring on a 1–5 scale. The rubric includes Accuracy, Trigger Specificity, Actionability, and Generalization:
>
> |Dimension|Text-Only|Multimodal|Delta|
> |---|---|---|---|
> |Accuracy|3.68|3.88|+0.20|
> |TrigSpec|3.77|4.12|+0.35|
> |Actionability|3.68|4.06|+0.38|
> |Generalization|3.66|3.74|+0.08|
>
> The advantage of the visual grounding setting stems from the precise specification of trigger conditions and the explicit detailing of action steps. It significantly reduces text heuristics that describe what to do without explaining how to do it, making the extracted knowledge more actionable.
>
> ### R5: Computational and Storage Costs for Long-Term Deployment
>
> > Question-4: What are the computational and storage costs of maintaining and retrieving from the hierarchical knowledge base?
>
> Storage is lightweight: the knowledge base is Markdown for Skills and JSON with text and embedding vectors for Experiences, and a full bank needs under 3 MB. Phase I maintenance is capped by the number of MLLM calls during each knowledge base update; with batch updates we average 1.1 calls per sample, and our integration mechanism limits total size so these costs do not grow exponentially with deployment time. Phase II retrieval adds little overhead: sub-query generation by the MLLM no more than 1024 tokens and one lightweight vector similarity search with text-embedding-3-small.
>
> We thank the reviewer for these questions; the clarifications and experiments strengthen the paper.

---

### Official Review · Reviewer_wK6u · 2026-03-13

**Soundness:** 3
**Presentation:** 2
**Significance:** 3
**Originality:** 2
**Overall Recommendation:** 4
**Confidence:** 3

**Summary:**

This paper introduces EXSKILL, a non-parametric continual learning framework designed for multimodal agents. The core idea is to externalize knowledge into two distinct representations: task-level Skills and action-level Experiences. These are subsequently accumulated, consolidated, retrieved, and adapted through a visually grounded pipeline. The authors evaluate EXSKILL across various multimodal agent benchmarks and model backbones, reporting consistent performance improvements over tool-only and prior memory-based baselines.

**Compliance With Llm Reviewing Policy:**

Affirmed.

**Final Justification:**

I believe the rebuttal addressed my major concerns and I will raise my recommendation to 4.

**Key Questions For Authors:**

1. Could you clarify the exact definition of an "independent run" in your reported results? Does this entail a complete rerun of both the accumulation and inference pipelines? Please provide the variance across these runs.

2. Which specific component of the framework should be considered the primary conceptual contribution: the dual-stream memory design, the visual grounding mechanism, or the adaptation step? Clarifying this would help properly evaluate the paper's novelty.

3. Can you better justify the use of the term "continual learning" given the current offline accumulation setup, or clarify how the evaluation reflects true continuous adaptation?

4. A direct empirical comparison between "text-only" and "visually grounded" knowledge extraction. This is essential to substantiate the paper's core claim regarding the absolute necessity of visual grounding.

5. Continual Learning Evaluation: An experiment demonstrating the agent's performance trajectory over a continuous, sequential stream of tasks to validate the continual learning claims.

If the author can solve above issues, this work will be much more solid.

**Limitations:**

yes

**Strengths And Weaknesses:**

Strength:
1. The paper tackles a persistent bottleneck in current multimodal agents: their episodic nature and inability to leverage past interactions. The motivation to move beyond text-only memory extraction is compelling and well-articulated.
2. The proposed method is coherent and intuitive. The dichotomy between Skills and Experiences is logically sound, and the visually grounded extraction and adaptation pipeline aligns perfectly with the requirements of multimodal tasks.
3. The experimental evaluation is commendable for its breadth. It encompasses multiple benchmarks, various LLM backbones, zero-shot transfer scenarios, and detailed behavioral ablations. The reported performance gains are substantial enough to warrant attention.

Weakness:
1. The characterization of the framework as "continual learning" is debatable. The experimental setup resembles an offline knowledge accumulation phase (from sampled training tasks) followed by an evaluation phase, rather than a true online continual learning paradigm where an agent's memory evolves dynamically over a continuous stream of tasks.
2. A central premise of the paper highlighted as a primary limitation of prior work is that existing methods rely on text-only trajectory logs and neglect visual modalities during knowledge extraction. consequently, "visually-grounded summarization" is presented as a core technical contribution. however, the ablation study (Table 3) completely lacks a direct empirical comparison to validate this claim. The authors ablate the entire Experience or Skill modules, but they do not isolate the visual modality variable. Without a direct "text-only extraction" vs. "visually-grounded extraction" comparison, the necessity and actual impact of the visual grounding in Phase I remain unproven.

---

> ### Author Rebuttal · Authors · 2026-03-31
>
> We thank the reviewer for constructive feedback and for recognizing our dual-stream design and broad experimental evaluation. New experiments and analyses may address your concerns and strengthen the paper's arguments.
>
> ### R1: Clarification and Experiments on Continual Learning
>
> > Weakness-1 & Question-3: Can you better justify the use of the term "continual learning" given the current offline accumulation setup?
>
> > Question-5: An experiment demonstrating the agent's performance trajectory over a continuous, sequential stream of tasks.
>
> While our main evaluation uses a train-eval split to demonstrate zero-shot transfer, EXSKILL is designed for online evolution where inference history continuously feeds back into the accumulation phase. To validate this, we test the agent on a continuous stream of tasks from VisualToolBench. Instead of a fixed knowledge base, the agent dynamically updates its memory during the task stream, making every updated experience or skill immediately available for subsequent tasks. We measure performance (Average@4) every 20 tasks to show how the agent adapts over time:
>
> |Accumulation Tasks|0|20|40|60|80|100|
> |---|---|---|---|---|---|---|
> |Gemini-2.5-Pro (%)|25.35|25.12|26.70|28.30|31.30|30.49|
> |Gemini-3-Flash (%)|41.94|42.76|44.86|44.28|45.79|46.50|
>
> Performance exhibits a clear upward trend as the agent processes more tasks and enriches its knowledge base. Unlike prompt optimizers like DSPy requiring full trajectory batches before offline extraction, EXSKILL updates dynamically. We will update the manuscript to articulate this online behavior and include these trajectory results.
>
> ### R2: Visual Grounding vs. Text-Only Extraction (Empirical Ablation)
>
> > Weakness-2: Without a direct "text-only extraction" vs. "visually-grounded extraction" comparison, the necessity and actual impact of the visual grounding remain unproven.
>
> > Question-4: A direct empirical comparison between "text-only" and "visually grounded" knowledge extraction.
>
> To validate the necessity of visual grounding, we introduce EXSKILL (Text-Only). In this setting, Phase I knowledge extraction relies strictly on text trajectory logs, and the MLLM is shielded from all initial and intermediate images during summarization and critique.
>
> Quantitative results on VisualToolBench (Gemini-2.5-Pro):
>
> |Setting|Avg@4|Delta|
> |---|---|---|
> |w/ Tools|25.35%|-|
> |EXSKILL(Text-Only)|27.57%|+2.22%|
> |EXSKILL(Full)|30.49%|+5.14%|
>
> While text-only extraction improves performance marginally, the full visual grounding pipeline delivers a much larger gain, confirming that visual observations are indispensable for capturing actionable tactical experiences.
>
> We also conduct a double-blind evaluation of extracted knowledge snippets using strong LLM judges (Gemini3-Pro). To ensure objectivity, reviewers see only de-identified skill and experience entries. The rubric covers accuracy, trigger specificity, actionability, and generalization:
>
> |Dimension|Text-Only|Full|Delta|
> |---|---|---|---|
> |Accuracy|3.68|3.88|+0.20|
> |TrigSpec|3.77|4.12|+0.35|
> |Actionability|3.68|4.06|+0.38|
> |Generalization|3.66|3.74|+0.08|
>
> Visual grounding avoids vague heuristics by capturing precise visual-semantic alignments. This empirical comparison, which we will add to Section 3.3, demonstrates the necessity of visual observations in Phase I.
>
> ### R3: Definition of "Independent Run"
>
> > Question-1: Could you clarify the exact definition of an "independent run"?
>
> An independent run refers strictly to the Phase II inference pipeline. For each test task, the agent solves from scratch independently $N=4$ times without re-running Phase I accumulation. The knowledge base stays fixed for that test set to ensure a fair comparison. The $N=4$ rollouts yield Average@4 (robustness) and Pass@4 (capability ceiling).
>
> ### R4: Clarification of Primary Conceptual Contribution
>
> > Question-2: Which specific component... should be considered the primary conceptual contribution?
>
> Our primary conceptual contribution is the dual-stream memory design, which bridges task-level Skills (procedural memory) and action-level Experiences (episodic memory). While prior agent frameworks often focus on a single memory type, EXSKILL integrates both to capture both high-level strategies and low-level execution details. This methodological innovation is particularly vital as agent skills transition from manual expert authoring to automatic, scalable generation.
>
> For multimodal agents, this dual-stream architecture is rendered effective through visual grounding and adaptation. Grounding ensures that extracted knowledge is anchored in actual visual states rather than just text logs, while adaptation enables the reuse of these insights on novel inputs. These components are not separate modules but form an integrated system designed specifically for vision-language tasks.
>
> We thank the reviewer for these questions; the clarifications and experiments strengthen the work.

---

> > ### Author Rebuttal · Reviewer_wK6u · 2026-04-04
> >
> > I will raise my recommendation to 4.

---

> > > ### Author Response · Authors · 2026-04-05
> > >
> > > Dear Reviewer wK6u,
> > >
> > > Thank you for reviewing our rebuttal and supporting our work! We noticed your note about raising the score to 4, but it seems the score hasn't been updated in the system yet. Could you please check and update it when convenient?
> > >
> > > If you have any further questions or points for discussion, please feel free to let us know. We'd be very happy to discuss them further. Thank you again for your time and insights!

---

### Official Review · Reviewer_wjSY · 2026-03-15

**Soundness:** 4
**Presentation:** 2
**Significance:** 3
**Originality:** 3
**Overall Recommendation:** 5
**Confidence:** 4

**Summary:**

EXSKILL is a framework for having multi-modal, tool-using LLM agents accumulate and reuse knowledge from previous attempts without changing parameters. This is done by creating two external repositories of knowledge for the model: Skills and experiences. Skills are task level guidance documents. Experiences are conditional tips where given a certain state, the LLM is advised to do some action. This is done in two phases. The first phase is abstraction of overall similarities and overlaps between tasks. The second phase leverages generated rollouts to extract patterns and populate the repository of skills and experiences. During inference, these skills and experiences are ranked via a similarity embedding, rewrites experiences to be relevant to the current task, and appends the rewritten experiences and skills into the prompt.

**Compliance With Llm Reviewing Policy:**

Affirmed.

**Final Justification:**

The authors answered all of my concerns with additional experiments and clarifications. I have raised my score accordingly (4 -> 5)

**Key Questions For Authors:**

My main question is just the point in the weaknesses about the lack of ablation over the text-only version. Why did the authors not consider it as it seems pretty central to their claim for contribution? To me, it feels like a prompt optimizer like DSPy would be a fairy natural baseline here.

**Limitations:**

The authors don't really engage with methodological limitations at all. Future work or weaknesses in the current approach just aren't really discussed in the conclusion. Pretty much everything I listed in the weaknesses feels like it begs some sort of acknowledgement in a short limitations discussion in the conclusion but there is just not a meaningful effort made in this direction.

**Strengths And Weaknesses:**

Strengths and Weaknesses:

(Authors can probably ignore the Significance/Soundness/Originality/Presentation +- notation, this is more leftovers from my notes that I figured might be helpful for other reviewers / the AC)

Strengths:
I think the biggest strength to me in terms of design is the experience and subsequent rewriting mechanism. Experiences alone might quickly grow too specific but the rewriting is a smart way of making them transferable. (Originality +, Soundness +)

Which leads to what I think is the biggest strength of the paper, which is that the generated knowledge base can simply be transplanted to another LLM without much issue. This is a very useful downstream and direct application. (Significance +)

The ablations are fairly comprehensive, Figure 2 and Tables 3 and 4 showing why both the skills and the experiences work in tandem. I think these ablations are very important, otherwise my immediate impression would have probably been ‘Why are skills different from claude skills, why is this needed as part of your framework?’ (Soundness +)

Not just the ablations but the main experiments themselves are fairly comprehensive, five benchmarks, three domains, and four models. These all strongly support the transferability argument. (Soundness +, Significance +)

Weaknesses:

There is a strong claim in the abstract: “These results demonstrate that our framework enables transferable continual learning for multimodal agents…” I see the transferable learning but not transferable continual learning. All of the experiments seem to be single-pass train-eval. (Presentation −)

There’s no ablation against the text-only version of this despite the vision aspect being heavily emphasized in the contributions. As part of this, the first immediate baseline that comes to mind would be prompt optimizers, ie DSPy. (Soundness −)

The skills are pretty much just like Claude skills but the authors acknowledge this so it’s not really a big deal in my mind. (Originality –)

---

> ### Author Rebuttal · Authors · 2026-03-31
>
> We thank the reviewer for recognizing our design's core strengths. We also appreciate the constructive feedback to enhance our manuscript.
>
> ### R1: Clarification on Continual Learning and Empirical Evidence
>
> > Weakness-1: I see the transferable learning but not transferable continual learning. All of the experiments seem to be single-pass train-eval.
>
> We clarify that EXSKILL Phase I is not a static offline process. In our framework, inference history continuously feeds back into the accumulation phase to guide knowledge extraction and entry modification. Every updated experience or skill becomes immediately available for subsequent tasks, enabling sustained performance gains. To demonstrate this dynamic behavior, we conduct a longitudinal experiment on VisualToolBench, evaluating agent performance (Average@4) every 20 tasks during the knowledge accumulation process:
>
> |Accumulation Tasks|0|20|40|60|80|100|
> |---|---|---|---|---|---|---|
> |Gemini-2.5-Pro (%)|25.35|25.12|26.70|28.30|31.30|30.49|
> |Gemini-3-Flash (%)|41.94|42.76|44.86|44.28|45.79|46.50|
>
> Performance exhibits a clear upward trend as the agent processes more tasks and enriches its knowledge base. Unlike prompt optimizers requiring full trajectory batches before offline extraction, EXSKILL updates dynamically. We will include this learning curve analysis in the revised appendix to substantiate our continual learning claim.
>
> ### R2: Ablation Study on Text-Only Version and DSPy Baselines
>
> > Weakness-2: There’s no ablation against the text-only version... such as prompt optimizers, i.e., DSPy.
>
> > Question-1: Why did the authors not consider it as it seems pretty central to their claim for contribution?
>
> Integrating visual observations is central to our contribution. To isolate visual grounding's impact, we introduce two new experiment sets on VisualToolBench using Gemini-2.5-Pro: (1) EXSKILL (Text-Only), where we ablate the visual modality in Phase I by relying strictly on text trajectory logs and shielding all images; (2) DSPy Baselines, evaluating text-only prompt optimization strategies BootstrapFewShot and MIPROv2 adapted to our tool-use setting.
>
> |Setting|Avg@4|Delta|
> |---|---|---|
> |w/ Tools|25.35%|-|
> |DSPy-BootstrapFewShot|24.42%|-0.93%|
> |DSPy-MIPROv2|25.82%|+0.47%|
> |EXSKILL(Text-Only)|27.57%|+2.22%|
> |EXSKILL(Full Multimodal)|30.49%|+5.14%|
>
> Text-only optimizers like DSPy struggle with complex tool-use sequences in multimodal tasks. While EXSKILL (Text-Only) performs better by using its dual-stream structure to organize knowledge from patterns in textual logs, EXSKILL (Full) still significantly outperforms all text-only baselines. This is because visual observations are essential for understanding the results of interleaved reasoning. For example, tool actions like cropping or rotating directly change how an object looks (e.g., checking if a 90-degree rotation aligns an object with a boundary). Without seeing these visual changes, text-only methods cannot learn the exact relationships between actions and their results in complex tasks.
>
> ### R3: Similarity to Claude Skills
>
> > Weakness-3: The skills are pretty much just like Claude skills but the authors acknowledge this so it’s not really a big deal in my mind.
>
> Our innovation lies not in the Skill markdown format but in how multimodal agents automatically extract these skills from visual trajectories and evolve them within continuous task streams. Most current skills are manually authored by human experts, which limits scalability.
>
> ### R4: Discussion on Methodological Limitations
>
> > Limitation: The authors don't really engage with methodological limitations at all... acknowledgement in a short limitations discussion.
>
> We agree with this point and will add a dedicated Limitations and Future Work section to the revised manuscript to discuss following limitations. First, effectiveness depends on the model's initial capability; it must complete a portion of tasks to extract valuable knowledge. Self-supervised extraction quality is constrained by the model's intrinsic level, meaning weaker models like Qwen3-VL struggle to benefit directly from EXSKILL. Second, a performance bottleneck exists when transferring knowledge to smaller models due to the capability gap, as shown in Appendix A. Future work will explore combining EXSKILL with reinforcement learning to enhance small models' knowledge extraction and utilization. Preliminary attempts in this field [1, 2] provide a reference for future directions.
>
> [1] Xia et al., SkillRL, arXiv:2602.08234 (2026).
>
> [2] Muhtar et al., Complementary RL, arXiv:2603.17621 (2026).
>
> Again, we thank the reviewer for the insightful comments which lead to a more solid paper.

---

> > ### Author Rebuttal · Reviewer_wjSY · 2026-04-02
> >
> > To the continual learning point, I was referring to when an entire trajectory for a task is completed, rather than reset the entire context, the next task simply continues using the same message history (and for this paper, the accumulated repository of skills and experiences). For clarification, does this mean the results are from a run across the benchmarks where the stored skills and experiences are accumulated across all tasks?
> >
> > As a follow up, was there any investigation into whether the order of the tasks significantly impacted performance?
> >
> > Otherwise, I'm happy with the additional experiments and clarifications made. I don't quite agree with the concerns raised by other reviewers in terms of novelty. I'm leaning towards moving up to a full accept but will reserve judgement until the end of the discussion period.

---

> > > ### Author Response · Authors · 2026-04-05
> > >
> > > We sincerely thank the reviewer for the response and positive assessment. Regarding your follow-up questions, we provide the following details:
> > >
> > > ### 1. Clarification on the Continual Learning Setup
> > >
> > > To clarify the experimental setup: the repository of skills and experiences is accumulated sequentially within each individual benchmark, rather than globally across all benchmarks. As different benchmarks encompass distinct domains, toolsets, and state spaces, maintaining a single global repository may induce significant "negative transfer" where experiences from one domain inappropriately interfere with another.
> > >
> > > Consequently, the continual learning process operates at the intra-benchmark level: Within a single benchmark, the agent processes the task stream sequentially. For each new task, the immediate message history is reset to prevent context window overflow. However, the agent utilizes and updates a shared, continuously evolving knowledge repository specific to that benchmark. This demonstrates intra-domain continual learning without reliance on long-context memorization.
> > >
> > > We also agree that the cross-benchmark knowledge accumulation and utilization suggested by the reviewer are highly valuable, particularly considering the progressive loading strategy of current agent SKILLs. We intend to enhance the EXSKILL paradigm in future work, for instance, by implementing a knowledge routing mechanism that loads the appropriate repository based on the current task. We believe this approach will improve robustness in complex, mixed task streams.
> > >
> > > ### 2. Impact of Task Order
> > >
> > > This is an insightful question. To investigate task order impact, we conducted a new experiment.We evaluated EXSKILL on VisualToolBench with Gemini-2.5-Pro. The 100 training tasks remained fixed while the input sequence was varied during the Phase I accumulation:
> > >
> > > 1. Default Order: The original randomized sequence (Seed 42) employed in the main experiments.
> > > 2. Reverse Order: The exact reverse of the default sequence.
> > > 3. Random Shuffle: A newly randomized sequence with a different seed (Seed 43).
> > >
> > > | Task Order | Average@4 (%) | Pass@4 (%) |
> > > | :--- | :--- | :--- |
> > > | Default Order | 30.49 | 46.73 |
> > > | Reverse Order | 31.31 | 45.32 |
> > > | Random Shuffle | 29.56 | 45.79 |
> > >
> > > As shown in the table, the performance variance across different task orders is marginal. This robustness is grounded in the knowledge extraction mechanism of EXSKILL, which avoids linear temporal memory logs by employing hierarchical consolidation. New experiences and skills are evaluated against the repository via semantic similarity and MLLM judgement. Highly similar entries merge into generalized statements, while others are appended, making the knowledge base an evolving semantic set. As the global task distribution remains constant, the repository converges to a similar generalized state, reducing interference from task permutations.
> > >
> > > While the framework demonstrates robustness to random task permutations, we consider curriculum sequence as a potential future direction. We hypothesize that an easy-to-hard task stream could accelerate the rate of knowledge convergence, allowing the agent to master foundational experience and skills before addressing complex edge cases.
> > >
> > > We will consider incorporating these discussions into the revised Appendix. We thank the reviewer for these follow-up questions, which help us strengthen the work.

---

### Decision · Program_Chairs · 2026-04-30

**Decision:**

Accept (regular)

**Comment:**

This paper introduces EXSKILL, a non-parametric continual learning framework for multimodal agents that accumulates reusable knowledge by separating task-level skills from action-level experiences. Reviewers praised the intuitive dual-stream memory architecture, the integration of visual context during knowledge extraction, and the empirical evaluations demonstrating transferability. Initial reviewer concerns focused on the validity of the "continual learning" framing, the lack of empirical ablations isolating visual grounding from text-only extraction, and the scalability and complexity of the proposed pipeline. The authors provided a rebuttal, supplying longitudinal experiments to validate the continual learning trajectory, a text-only ablation that the necessity of visual grounding, and a double-blind LLM-judge evaluation confirming the quality and actionability of the extracted knowledge. The reviewers agreed that the rebuttal resolved their concerns, leading to a consensus of positive recommendations (one Accept and three Weak Accepts, with one reviewer noting their intent to raise score post-rebuttal). Given the methodological contribution to multimodal agent memory, the experimental validation, and the author response, the paper is recommended for acceptance.